# On the Diversity of Adversarial Ensemble Learning

**Jun-Qi Guo** [* 1]   **Meng-Zhang Qian** [* 1]   **Wei Gao** [1]   **Zhi-Hua Zhou** [1]

## Abstract

Diversity has been one of the most crucial factors on the design of adversarial ensemble methods. This work focuses on the fundamental problems: How to define the diversity for the adversarial ensemble, and how to correlate with algorithmic performance. We first show that it is an NP-Hard problem to precisely calculate the diversity of two networks in adversarial ensemble learning, which makes it different from prior diversity analysis. We present the first diversity decomposition under the first-order approximation for the adversarial ensemble learning. Specifically, the adversarial ensemble loss can be decomposed into average of individual adversarial losses, *gradient diversity*, *prediction diversity* and *cross diversity*. Hence, it is not sufficient to merely consider the gradient diversity on the characterization of diversity as in previous adversarial ensemble methods. We present diversity decomposition for classification with cross-entropy loss similarly. Based on the theoretical analysis, we develop new ensemble method via orthogonal adversarial predictions to simultaneously improve gradient diversity and cross diversity. We finally conduct experiments to validate the effectiveness of our method.

## 1. Introduction

General machine learning models may be misled heavily by examples with adversarial perturbations (Szegedy et al., 2014; Goodfellow et al., 2015), which raises some serious concerns about reliability of models, particularly in high-risk applications such as healthcare, finance and autonomous driving (Finlayson et al., 2019; Deng et al., 2021; Fursov et al., 2021). Various robust methods have been developed against adversarial examples in recent years (Zhang et al.,

2019b; Gowal et al., 2021; Pang et al., 2021; Wang et al., 2023; Peng et al., 2023; Bartoldson et al., 2024).

Ensemble learning combines multiple learners rather than one single learner with better performance, which has been paid much attention in the adversarial robustness learning. Sequential robust ensemble has been constructed via the boosting framework (Abernethy et al., 2021; Zhang et al., 2019a; 2022; Guo et al., 2022), and parallel ensemble has also developed for robust learning (Pinot et al., 2020; Sen et al., 2020; Yang et al., 2021; 2022; Deng & Mu, 2024).

Diversity has always been one of the most crucial factors in the design of ensemble methods (Zhou, 2012; Wood et al., 2023). For adversarial ensemble learning, Pang et al. (2019) took prediction disagreements of base learners as diversity, while Yang et al. (2020) defined the diversity via losses of base learners on exchanged adversarial examples. Another idea is to consider the misalignment of gradient directions for diversity (Kariyappa & Qureshi, 2019; Dabouei et al., 2020; Huang et al., 2021; Bogun et al., 2022).

There are some fundamental problems open for adversarial ensemble learning. For example, how to formally define the diversity of adversarial ensemble, and what's more, how to correlate diversity definition with algorithmic performance from a theoretical view. This work studies fundamental problems of diversity in the adversarial ensemble learning, and the main contributions are summarized as follows:

- We first show that it is an NP-Hard problem to precisely calculate the diversity of two neural networks in the adversarial ensemble, since diversity is heavily relevant to intrinsic structures and output predictions of models simultaneously. This challenge makes it different from traditional diversity on output predictions (Zhou, 2012; Wood et al., 2023). Sun & Zhou (2018) indicated the importance of structure diversity for decision trees, whereas it remains open for neural networks.

- We present the first diversity decomposition under the first-order approximation in the adversarial ensemble learning. Specifically, the adversarial ensemble loss is decomposed into average of individual adversarial losses, *prediction diversity*, *gradient diversity* and *cross diversity*. It is not sufficient to only consider gradient diversity on the characterization of diversity as in prior

---

*Equal contribution [1]National Key Laboratory for Novel Software Technology, Nanjing University, Nanjing, China; School of Artificial Intelligence, Nanjing University, Nanjing, China. Correspondence to: Wei Gao <gaow@lamda.nju.edu.cn>.

*Proceedings of the $42^{nd}$ International Conference on Machine Learning*, Vancouver, Canada. PMLR 267, 2025. Copyright 2025 by the author(s).

adversarial ensemble methods. We present similar decomposition for classification with cross-entropy loss, which is commonly used for neural networks.

- Based on theoretical analysis, we develop the AdvE$_{\text{OAP}}$ adversarial ensemble method[1] via the orthogonal of adversarial predictions of base learners, which could improve gradient and cross diversity simultaneously. We finally conduct empirical studies to validate the effectiveness of our AdvE$_{\text{OAP}}$ method.

The rest of this work is constructed as follows: Section 2 presents some preliminaries. Section 3 provides diversity analysis. Section 4 develops our method. Section 5 conducts experiments. Section 6 concludes with future work.

## 2. Preliminary

Let $\mathcal{X} \subseteq \mathbb{R}^d$ and $\mathcal{Y}$ denote the instance and label space, respectively, where $\mathcal{Y} = \{0, 1\}$ for binary classification and $\mathcal{Y} \subseteq R$ for regression. Let $\mathcal{D}$ be an underlying distribution over the product space $\mathcal{X} \times \mathcal{Y}$, and we have a training data

$$S_n = \{(\boldsymbol{x}_1, y_1), (\boldsymbol{x}_2, y_2), \cdots, (\boldsymbol{x}_n, y_n)\},$$

where each sample is drawn i.i.d. from distribution $\mathcal{D}$.

Define the perturbation set $\Delta_p^\epsilon$ w.r.t. $l_p$ norm and $\epsilon > 0$ as

$$\Delta_p^\epsilon = \{\boldsymbol{\delta} \in \mathbb{R}^d \colon \|\boldsymbol{\delta}\|_p = (|\delta_1|^p + |\delta_2|^p + \cdots + |\delta_d|^p)^{1/p} \le \epsilon\},$$

which shows instances' imperceptible perturbation (Szegedy et al., 2014). Let $\mathcal{F} = \{f : \mathcal{X} \to \mathbb{R}\}$ be a hypothesis space, and loss function $\ell : \mathcal{Y} \times \mathcal{Y} \to \mathbb{R}$ is introduced to measure performance. Given $f \in \mathcal{F}$, we define the adversarial perturbation w.r.t. example $\boldsymbol{x}$ as

$$\boldsymbol{\delta}_f^* \in \arg\max_{\boldsymbol{\delta} \in \Delta_p^\epsilon} \ell(f(\boldsymbol{x} + \boldsymbol{\delta}), y),$$

and $\boldsymbol{x} + \boldsymbol{\delta}_f^*$ is called the adversarial example w.r.t. $\boldsymbol{x}$.

We define the expected adversarial loss as

$$\mathcal{L}^{\text{adv}}(f, \mathcal{D}) = \mathbb{E}_{(\boldsymbol{x}, y) \sim \mathcal{D}} \Big[ \max_{\boldsymbol{\delta} \in \Delta_p^\epsilon} \{\ell(f(\boldsymbol{x} + \boldsymbol{\delta}), y)\} \Big],$$

and define the empirical adversarial loss w.r.t. data $S_n$ as

$$\hat{\mathcal{L}}^{\text{adv}}(f, S_n) = \frac{1}{n} \sum_{i=1}^{n} \max_{\boldsymbol{\delta} \in \Delta_p^\epsilon} \{\ell(f(\boldsymbol{x}_i + \boldsymbol{\delta}), y_i)\}.$$

We finally introduce some useful notations in this work. Denote by $\langle \cdot, \cdot \rangle$ the inner product of two vectors, and $\boldsymbol{e}_i$ is a unit vector with $i$-th element 1. Write $[k] = \{1, 2, \cdots, k\}$

[1] Code is available at https://github.com/GuoJQ42/AdvOAP.

for integer $k > 0$. For two non-negative real numbers $a$ and $b$ with $a + b = 1$, we define the KL-divergence as

$$\text{KL}(a, b) = a \ln(a/b) + (1 - a) \ln((1 - a)/(1 - b)),$$

and for two probability vectors $\boldsymbol{a} = (a_1, a_2, \cdots, a_m)$ and $\boldsymbol{b} = (b_1, b_2, \cdots, b_m)$, we define the KL-divergence as

$$\text{KL}(\boldsymbol{a}, \boldsymbol{b}) = \sum_{i=1}^{m} a_i \ln(a_i/b_i).$$

## 3. Theoretical Analysis on Diversity

Given $m$ learners $f_1(\boldsymbol{x}), ..., f_m(\boldsymbol{x})$, this work focuses on the simplest ensemble method (Dietterich, 2000; Zhou, 2012)

$$\bar{f}(\boldsymbol{x}) = \sum_{j=1}^{m} \frac{f_j(\boldsymbol{x})}{m}.$$

We begin with the challenge on the analysis of adversarial diversity, and then present diversity decomposition w.r.t. squared loss and cross-entropy loss, respectively.

### 3.1. Main challenge on analysis of adversarial diversity

For traditional non-adversarial ensemble learning, it is easy to make the error-ambiguity decomposition over example $(\boldsymbol{x}, y)$ w.r.t. squared loss from (Krogh & Vedelsby, 1994; Zhou, 2012) as follows:

$$\underbrace{(\bar{f}(\boldsymbol{x}) - y)^2}_{\text{ensemble loss}} = \underbrace{\sum_{j=1}^{m} \frac{(f_j(\boldsymbol{x}) - y)^2}{m}}_{\text{average loss}} - \underbrace{\sum_{j=1}^{m} \frac{(f_j(\boldsymbol{x}) - \bar{f}(\boldsymbol{x}))^2}{m}}_{\text{ambiguity}},$$

where the ambiguity can be viewed as ensemble diversity. Wood et al. (2023) further presented bias-variance-diversity decomposition for ensemble learning, simplified by

$$\textbf{ensemble loss} = \textbf{bias} + \textbf{variance} - \textbf{diversity}.$$

It is natural to consider some similar decompositions in the adversarial diversity learning. However, this remains some challenges as shown by the following theorem.

**Theorem 3.1.** *For squared loss, it is an NP-hard problem to precisely calculate the diversity w.r.t. example $(\boldsymbol{x}, y)$ in the adversarial ensemble learning as follows:*

$$\frac{1}{2} \sum_{j=1}^{2} \max_{\boldsymbol{\delta} \in \Delta_p^\epsilon} \{(f_j(\boldsymbol{x} + \boldsymbol{\delta}) - y)^2\} - \max_{\boldsymbol{\delta} \in \Delta_p^\epsilon} \{(\bar{f}(\boldsymbol{x} + \boldsymbol{\delta}) - y)^2\},$$

*where the error ambiguity decomposition is considered for two neural networks $f_1(\boldsymbol{x})$ and $f_2(\boldsymbol{x})$ with ReLU activation, and $\bar{f}(\boldsymbol{x}) = (f_1(\boldsymbol{x}) + f_2(\boldsymbol{x}))/2$.*

Theorem 3.1 shows that, even for the ensemble of two neural networks, it is an NP-hard problem to make error ambiguity decomposition in the adversarial ensemble learning. The main challenge is that the adversarial example is heavily dependent on the intrinsic structure of neural network, and diversity analysis is relevant to intrinsic structure, model prediction and adversarial perturbation simultaneously. This is different from traditional diversity analysis on model predictions (Zhou, 2012; Wood et al., 2023). We also notice the importance of structure diversity for decision trees (Sun & Zhou, 2018), whereas it remains open for neural networks.

The proof idea involves the reduction of 3-SAT problem. Specially, we consider a 3-SAT problem with $k$ clauses, and each clause is a disjunction of 3 literals. We construct a neural network $g(\boldsymbol{x})$ with $\Theta(k)$ layers and $\Theta(k)$ width, where each literal can be regarded as an input node of $g(\boldsymbol{x})$, and each disjunction and conjunction can be replaced with the $\max$ and $\min$ operators, respectively. The $\max$ and $\min$ operators can be constructed by ReLU activation. We construct two neural networks $f_1(\boldsymbol{x}, x_0) = \min(g(\boldsymbol{x}), x_0)$ and $f_2(\boldsymbol{x}, x_0) = \min(g(\boldsymbol{x}), -x_0)$ by adding an auxiliary variable $x_0$. The detailed proof is given in Appendix A, which is partially motivated from previous work on $l_1$ or $l_\infty$ norm (Katz et al., 2017; Weng et al., 2018), while our work generalizes to $l_p$ norm with $p = 1, 2, \cdots, \infty$.

## 3.2. Diversity decomposition w.r.t. squared loss

Previous ensemble methods generally take the first-order approximation for adversarial loss function, and focus on gradient diversity (Kariyappa & Qureshi, 2019; Dabouei et al., 2020; Huang et al., 2021). This is partially because of the NP-hardness for adversarial diversity in Theorem 3.1. Following the first-order Taylor approximation, we have

$$\bar{f}(\boldsymbol{x}+\boldsymbol{\delta}) = \sum_{j=1}^{m} \frac{f_j(\boldsymbol{x}+\boldsymbol{\delta})}{m} \approx \sum_{j=1}^{m} \frac{f_j(\boldsymbol{x})}{m} + \sum_{j=1}^{m} \frac{\nabla f_j(\boldsymbol{x})^T \boldsymbol{\delta}}{m} .$$

We now present the first decomposition for the adversarial ensemble loss w.r.t. squared loss as follows:

**Theorem 3.2.** *For ensemble $\bar{f} = \sum_{j=1}^{m} f_j/m$, we present the decomposition of adversarial ensemble loss under the first-order approximation over example $(\boldsymbol{x}, y)$ as follows:*

$$\max_{\boldsymbol{\delta}\in\Delta_p^\epsilon}\{(\bar{f}(\boldsymbol{x}+\boldsymbol{\delta})-y)^2\} = \underbrace{\sum_{j=1}^{m} \frac{\max_{\boldsymbol{\delta}\in\Delta_p^\epsilon}\{(f_j(\boldsymbol{x}+\boldsymbol{\delta})-y)^2\}}{m}}_{\textit{average of individual adversarial losses}}$$

$$\underbrace{-\sum_{j=1}^{m} \frac{(f_j(\boldsymbol{x})-\bar{f}(\boldsymbol{x}))^2}{m}}_{\textit{prediction diversity}} \underbrace{-\epsilon^2 \sum_{j=1}^{m} \frac{\|\nabla f_j(\boldsymbol{x})\|_q^2 - \|\nabla \bar{f}(\boldsymbol{x})\|_q^2}{m}}_{\textit{gradient diversity}}$$

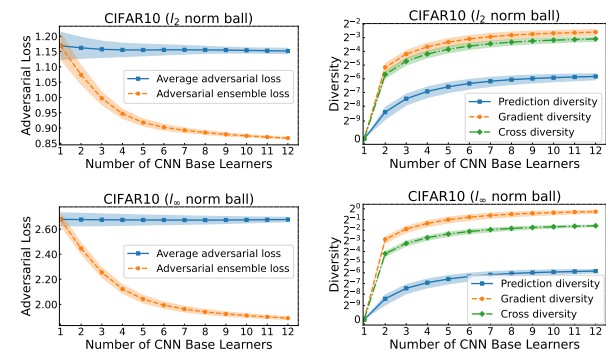

Figure 1. An illustration of diversity decomposition in Theorem 3.2 on dataset CIFAR10 of perturbation balls with $l_2$ and $l_\infty$ norm, respectively. Here, we consider CNNs as base learners.

$$\underbrace{-2\epsilon \sum_{j=1}^{m} \frac{\|\nabla f_j(\boldsymbol{x})\|_q |f_j(\boldsymbol{x}) - y| - \|\nabla \bar{f}(\boldsymbol{x})\|_q |\bar{f}(\boldsymbol{x}) - y|}{m}}_{\textit{cross diversity}}$$

*where* $1/p + 1/q = 1$.

In this theorem, the adversarial ensemble loss is decomposed into the average of individual adversarial losses, *prediction diversity*, *gradient diversity* and *cross diversity*. Gradient diversity measures the dispersion of gradients concerning the mean of gradients w.r.t. $l_q$ norm, and it exactly becomes the variance as for $q = 2$. Prediction diversity can be understood as the traditional diversity such as ambiguity (Krogh & Vedelsby, 1994; Zhou, 2012). Cross diversity can be viewed as a cross between functional outputs and gradients. Gradient and cross diversities are highly relevant to intrinsic structures of functions and functional outputs. The detailed proof is given in Appendix B.1.

From Theorem 3.2, it is also observable that gradient and cross diversities take more important roles on adversarial ensemble learning as for larger $\epsilon$ (i.e., radius of perturbation ball), yet prediction diversity dominates as for smaller $\epsilon$. Also, the decomposition is relevant to the $l_p$-norm distance of perturbation ball. Thus, the characterization of diversities in adversarial ensemble learning is more complicated than that of traditional non-adversarial ensemble learning.

Figure 1 presents an intuitive illustration for the diversity decomposition of Theorem 3.2. Here, we consider dataset CIFAR10 with two classes, and focus on the perturbation ball with the popular $l_2$ and $l_\infty$ norm. More experiment details are given in Appendix B.2, and we try to understand the trends of loss functions and diversities as the number of CNN-base learners increases.

As can be seen from Figure 1, we have smaller adversarial ensemble loss as for larger prediction, gradient and cross diversities w.r.t. $l_2$ and $l_\infty$ norm perturbation balls, when we

keep average of individual adversarial losses stable. Generally, gradient and cross diversities are larger than prediction diversity and take more important roles. This is nicely in accordance with our Theorem 3.2, and diversity empirically plays an important role in adversarial ensemble learning.

We focus on the first-order approximation as in previous adversarial ensemble methods (Kariyappa & Qureshi, 2019; Dabouei et al., 2020; Huang et al., 2021), and it is interesting to explore the second-order (or higher-order) approximation from gradients and Hessian matrices. The main challenge is how to obtain the closed-form solution of adversarial loss via second-order approximation, because it is relevant to the roots of high-order polynomials (Forsythe & Golub, 1965; Moré & Sorensen, 1983; Fortin & Wolkowicz, 2004). More discussions on this issue are given in Appendix B.3.

**Relevant to previous adversarial ensemble methods**

Most previous ensemble methods consider the first-order approximation of adversarial loss functions (Kariyappa & Qureshi, 2019; Dabouei et al., 2020; Yang et al., 2021; Bogun et al., 2022), and the diversity is measured by the average of $\cos$ values over pairs of gradients of base learner. Specifically, the diversity w.r.t. instance $\boldsymbol{x}$ is given by

$$\frac{1}{m(m-1)} \sum_{i=1}^{m} \sum_{j=i+1}^{m} \cos(\nabla f_i(\boldsymbol{x}), \nabla f_j(\boldsymbol{x})),$$

where $\cos(\nabla f_i(\boldsymbol{x}), \nabla f_j(\boldsymbol{x}))$ denotes the $\cos$ value of the angle between $\nabla f_i(\boldsymbol{x})$ and $\nabla f_j(\boldsymbol{x})$. We could derive the relationship between our gradient diversity and previous diversity via the $\cos$ functions as follows:

$$\text{Gradient diversity} = \frac{m\epsilon^2 - \epsilon^2}{m^2} \sum_{i=1}^{m} \|\nabla f_i(\boldsymbol{x})\|_2^2$$

$$-\frac{\epsilon^2}{m^2} \sum_{i \neq j} \|\nabla f_i(\boldsymbol{x})\|_2 \|\nabla f_j(\boldsymbol{x})\|_2 \cos(\nabla f_i(\boldsymbol{x}), \nabla f_j(\boldsymbol{x})) \,.$$

It is feasible to enlarge gradient diversity by decreasing $\cos$ functions, which is nicely in accordance with previous work (Dabouei et al., 2020; Bogun et al., 2022). Meanwhile, it is noteworthy of other important factors on gradient diversity such as gradient norm, rather than only one factor, which can be shown by following examples.

*Example* 1. There exist two ensembles of the same averages of $\cos$ values and individual adversarial losses, but with different adversarial ensemble losses.

*Proof.* We focus on 2-dimensional instance space $\mathcal{X} \subseteq \mathbb{R}^2$ and label space $\mathcal{Y} \subseteq \mathbb{R}$, and consider

$$f_1(\boldsymbol{x}) = x_1 + x_2, \qquad f_2(\boldsymbol{x}) = x_1 - 3x_2 \,,$$
$$f_3(\boldsymbol{x}) = abx_1 + bx_2, \quad f_4(\boldsymbol{x}) = abx_1 - bx_2 \,,$$

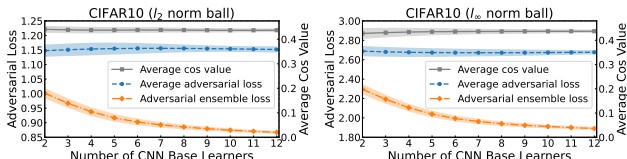

*Figure 2.* An illustration of the influence of average $\cos$ value on dataset CIFAR10 of perturbation balls with $l_2$ and $l_\infty$ norm, respectively. Here, we consider CNNs as base learners.

where $a = (\sqrt{5}-1)/2$ and $b = (\sqrt{6}+1)/(a+\sqrt{\sqrt{5}a})$. We study two ensembles: one ensemble of $f_1$ and $f_2$; the other ensemble of $f_3$ and $f_4$. For example $(\boldsymbol{x}, y) = ([1, 0], 1)$ and perturbation set $\Delta = \{\boldsymbol{\delta} : \|\boldsymbol{\delta}\|_2 \leq 2\}$, we have

$$\cos(\nabla f_1(\boldsymbol{x}), \nabla f_2(\boldsymbol{x})) = \cos(\nabla f_3(\boldsymbol{x}), \nabla f_4(\boldsymbol{x})) \,,$$

also with the same average of individual adversarial losses

$$\sum_{i=1}^{2} \max_{\delta \in \Delta} \frac{(f_i(\boldsymbol{x} + \boldsymbol{\delta}) - y)^2}{2} = \sum_{i=3}^{4} \max_{\delta \in \Delta} \frac{(f_i(\boldsymbol{x} + \boldsymbol{\delta}) - y)^2}{2} \,.$$

However, the adversarial ensemble losses are different from

$$\max_{\delta \in \Delta}(f_1(\boldsymbol{x} + \boldsymbol{\delta})/2 + f_2(\boldsymbol{x} + \boldsymbol{\delta})/2 - y)^2 = 8 \,,$$
$$\max_{\delta \in \Delta}(f_3(\boldsymbol{x} + \boldsymbol{\delta})/2 + f_4(\boldsymbol{x} + \boldsymbol{\delta})/2 - y)^2 \approx 7.2 \,.$$

Here, we consider the $l_2$ norm in perturbation set $\Delta$, and similar analysis could be made for $l_p$-norm. More details are presented in Appendix B.4. ∎

In addition to Example 1, we could also present empirical studies on adversarial ensemble losses versus the $\cos$ values over dataset CIFAR10, and the experiment details are given in Appendix B.2. Figure 2 shows the curves of adversarial ensemble loss, the averages of $\cos$ values and individual adversarial losses with $l_2$ and $l_\infty$ norm.

From Figure 2, it is clear that adversarial ensemble losses keep decreasing when we increase number of CNN-base learners, whereas it almost remains constant for the averages of $\cos$ values and individual adversarial losses. Therefore, it is not sufficient to characterize the diversity of adversarial ensemble by merely considering the average of $\cos$ values as in (Dabouei et al., 2020; Bogun et al., 2022).

### 3.3. Diversity decomposition w.r.t. cross-entropy loss

We study the decomposition of adversarial ensemble w.r.t. cross-entropy loss for binary classification, and also follow the first-order approximation $f(\boldsymbol{x} + \boldsymbol{\delta}) \approx f(\boldsymbol{x}) + \nabla f(\boldsymbol{x})^T \boldsymbol{\delta}$. For base learners with logit outputs (i.e., the logarithm of the ratio of probabilities), we have the probability of the positive class over $\boldsymbol{x} + \boldsymbol{\delta}$ as follows:

$$p_f(\boldsymbol{x} + \boldsymbol{\delta}) = \left(1 + \exp(-(f(\boldsymbol{x}) + \nabla f(\boldsymbol{x})^T \boldsymbol{\delta}))\right)^{-1} \,. \quad (1)$$

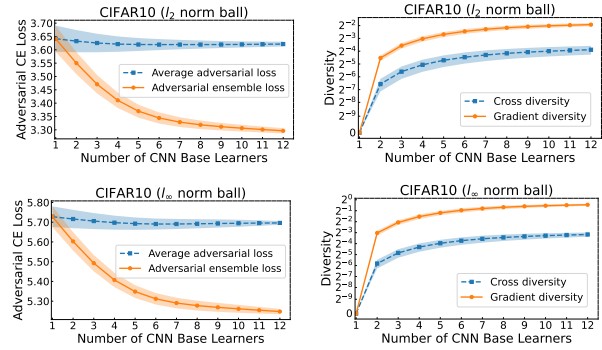

*Figure 3.* An illustration of diversity decomposition in Theorem 3.4 on dataset CIFAR10 of perturbation balls with $l_2$ and $l_\infty$ norm, respectively. Here, we consider CNNs as base learners.

The cross-entropy loss over $(\boldsymbol{x} + \boldsymbol{\delta}, y)$ is given by

$$
\ell(f(\boldsymbol{x} + \boldsymbol{\delta}), y) = -y(f(\boldsymbol{x}) + \nabla f(\boldsymbol{x})^T \boldsymbol{\delta}) \\
+ \ln(1 + \exp(f(\boldsymbol{x}) + \nabla f(\boldsymbol{x})^T \boldsymbol{\delta})) ,
$$

as in (Bishop & Nasrabadi, 2006), and recall that

$$
\boldsymbol{\delta}_f^* \in \arg\max_{\boldsymbol{\delta} \in \Delta_p^\epsilon} \ell(f(\boldsymbol{x} + \boldsymbol{\delta}), y) . \tag{2}
$$

We have the following closed-form solution for $p_f(\boldsymbol{x} + \boldsymbol{\delta}^*)$, and the detailed proof is presented in Appendix C.1.

**Lemma 3.3.** *For function $f$ and example $(\boldsymbol{x}, y)$, we have*

$$
p_f(\boldsymbol{x} + \boldsymbol{\delta}_f^*) = \left(1 + \exp(-(f(\boldsymbol{x}) - (2y-1)\|\nabla f(\boldsymbol{x})\|_q \epsilon))\right)^{-1}
$$

*where probability of positive class $p_f(\cdot)$ and perturbation $\boldsymbol{\delta}_f^*$ are defined by Eqns. (1) and (2), respectively.*

Based on this lemma, we have

**Theorem 3.4.** *For ensemble $\bar{f} = \sum_{j=1}^m f_j / m$, we have the decomposition of adversarial ensemble loss under the first-order approximation over example $(\boldsymbol{x}, y)$ as follows:*

$$
\max_{\boldsymbol{\delta} \in \Delta_p^\epsilon} \ell(\bar{f}(\boldsymbol{x} + \boldsymbol{\delta}), y) = \underbrace{\sum_{j=1}^m \frac{\max_{\boldsymbol{\delta} \in \Delta_p^\epsilon} \ell(f_j(\boldsymbol{x} + \boldsymbol{\delta}), y)}{m}}_{\text{average of individual adversarial losses}}
$$

$$
-r \underbrace{\sum_{j=1}^m \frac{\|\nabla f_j(\boldsymbol{x})\|_q - \|\nabla \bar{f}(\boldsymbol{x})\|_q}{m}}_{\text{gradient diversity}} - \underbrace{\sum_{j=1}^m \frac{KL(p_{\bar{f}}(\tilde{\boldsymbol{x}}_{\bar{f}}), p_{f_j}(\tilde{\boldsymbol{x}}_{f_j}))}{m}}_{\text{cross diversity}}
$$

*where $1/p + 1/q = 1$, $r = \epsilon(y - p_{\bar{f}}(\tilde{\boldsymbol{x}}_{\bar{f}}))(2y-1)$, $p_f(\cdot)$ is defined by Eqn. (1), and $\tilde{\boldsymbol{x}}_f = \boldsymbol{x} + \boldsymbol{\delta}_f^*$ is the adversarial example with $\boldsymbol{\delta}_f^*$ from Eqn. (2).*

In this theorem, the adversarial ensemble loss is decomposed into the average of individual adversarial losses, *gradient*

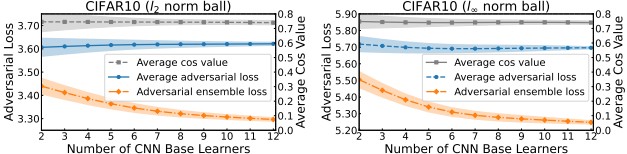

*Figure 4.* An illustration of influence of the average cos value on dataset CIFAR10 of perturbation balls w.r.t. $l_2$ and $l_\infty$ norm, respectively. Here, we consider CNNs as base learners.

*diversity* and *cross diversity*. The cross diversity shows the diversity of base learners according to their KL divergence to ensemble $\bar{f}$, which is relevant to function outputs and gradients, as shown in Lemma 3.3. It is evident that the decomposition of Theorem 3.4 is quite different from that of Theorem 3.2 because of different loss functions. The detailed proof is presented in Appendix. C.2.

We also present Figure 3 to illustrate the decomposition of Theorem 3.4, and some experimental details are given in Appendix B.2. As can be seen, adversarial ensemble loss gets smaller as for larger cross and gradient diversities w.r.t. $l_2$ and $l_\infty$ norm perturbation balls, when we maintain the average of individual adversarial losses stable. In addition, it is not sufficient to characterize the diversity of adversarial ensemble by only merely considering the average of cos values as done in (Dabouei et al., 2020; Bogun et al., 2022). We also present some empirical studies on the influence of average cos values on adversarial ensemble loss shown in Figure 4, and more details are given in Appendix C.3.

## 4. Our AdvE$_{\text{OAP}}$ Method

Motivated from our theoretical analysis in Theorem 3.4, we develop a robust ensemble method for multi-class learning, and the basic idea is to train multiple deep neural networks for adversarial ensemble via a regularization by considering gradient diversity and cross diversity simultaneously.

For multi-class learning with $\kappa$ classes, the base learner $\boldsymbol{f} = (f_1, f_2, \cdots, f_\kappa) \colon \mathcal{X} \to \mathbb{R}^\kappa$ maps each instance to a $\kappa$-dimensional logit vector. The predicted probability vector $\boldsymbol{p_f}(\boldsymbol{x}) = (p_{f_1}(\boldsymbol{x}), p_{f_2}(\boldsymbol{x}), \cdots, p_{f_\kappa}(\boldsymbol{x}))$ can be calculated from $\boldsymbol{f}(\boldsymbol{x})$ via softmax function as follows

$$
p_{f_k}(\boldsymbol{x}) = \frac{\exp(f_k(\boldsymbol{x}))}{\sum_{j=1}^\kappa \exp(f_j(\boldsymbol{x}))} . \tag{3}
$$

This section also focuses on the first-order approximation $\boldsymbol{f}(\boldsymbol{x} + \boldsymbol{\delta}) \approx \boldsymbol{f}(\boldsymbol{x}) + \boldsymbol{J_f}(\boldsymbol{x})\boldsymbol{\delta}$, where $\boldsymbol{J_f}(\boldsymbol{x})$ is the Jacobi matrix of $\boldsymbol{f}$ (Rudin et al., 1964). Denote by

$$
\boldsymbol{\delta_f}^* \in \arg\max_{\boldsymbol{\delta} \in \Delta_p^\epsilon} \ell(\boldsymbol{f}(\boldsymbol{x} + \boldsymbol{\delta}), y) . \tag{4}
$$

For $m$ base learners $\boldsymbol{f}_1, \cdots, \boldsymbol{f}_m$ with $\boldsymbol{f}_j = (f_{j,1}, \cdots, f_{j,\kappa})$,

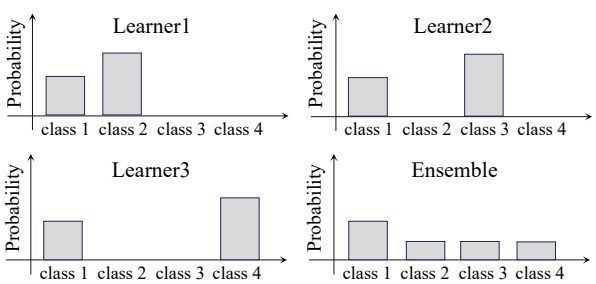

*Figure 5.* An illustration of orthogonality by optimizing Eqn. (5).

we have cross diversity for multi-classification

$$\text{Cross diversity} = \sum_{j=1}^{m} \frac{\text{KL}(\boldsymbol{p}_{\bar{\boldsymbol{f}}}(\boldsymbol{x} + \boldsymbol{\delta}_{\bar{\boldsymbol{f}}}^{*}), \boldsymbol{p}_{\boldsymbol{f}_{j}}(\boldsymbol{x} + \boldsymbol{\delta}_{\boldsymbol{f}_{j}}^{*}))}{m} ,$$

where $\boldsymbol{p}_{\boldsymbol{f}}(\cdot)$ and $\boldsymbol{\delta}_{\boldsymbol{f}}^{*}$ are given by Eqns. (3)-(4), respectively. From some algebraic derivations in Appendix D.1, we have

$$\begin{aligned}\text{Gradient diversity} &= \sum_{j=1}^{m} \frac{\langle \boldsymbol{r}, \boldsymbol{J}_{\boldsymbol{f}_{j}}(\boldsymbol{x})\boldsymbol{\delta}_{\boldsymbol{f}_{j}}^{*} - \boldsymbol{J}_{\bar{\boldsymbol{f}}}(\boldsymbol{x})\boldsymbol{\delta}_{\bar{\boldsymbol{f}}}^{*}\rangle}{m} \\ &= \sum_{j=1}^{m} \frac{\langle \boldsymbol{r}, \boldsymbol{f}_{j}(\boldsymbol{x} + \boldsymbol{\delta}_{\boldsymbol{f}_{j}}^{*}) - \bar{\boldsymbol{f}}(\boldsymbol{x} + \boldsymbol{\delta}_{\bar{\boldsymbol{f}}}^{*})\rangle}{m}\end{aligned}$$

where $\boldsymbol{r} = \boldsymbol{p}_{\bar{\boldsymbol{f}}}(\boldsymbol{x} + \boldsymbol{\delta}_{\bar{\boldsymbol{f}}}^{*}) - \boldsymbol{e}_{y}$.

For multi-class learning, we could simultaneously improve cross diversity and gradient diversity by diversifying the output predictions of adversarial examples of base learners. This motivates us to develop a new ensemble algorithm via orthogonal adversarial predictions, i.e., we orthogonalize the outputs over adversarial examples for base learners to improve diversity. The orthogonal idea is inspired by (Pang et al., 2019), which is limited only on clean examples, while our work generalizes to adversarial examples.

Specifically, we introduce a regularization for orthogonal adversarial output predictions of $\boldsymbol{f}_{1}, \cdots, \boldsymbol{f}_{m}$ as

$$\Gamma_{\alpha}(\boldsymbol{x}, y) = H\left(\sum_{j=1}^{m} \frac{\tilde{\boldsymbol{p}}_{\boldsymbol{f}_{j}}(\boldsymbol{x} + \boldsymbol{\delta}_{\boldsymbol{f}_{j}}^{*})}{m}\right) + \alpha \log(\det(\mathbf{A}^{T}\mathbf{A})) ,$$

where $H(\cdot)$ is the function of information entropy and

$$\mathbf{A} = [\tilde{\boldsymbol{p}}_{\boldsymbol{f}_{1}}(\boldsymbol{x} + \boldsymbol{\delta}_{\boldsymbol{f}_{1}}^{*}), \cdots, \tilde{\boldsymbol{p}}_{\boldsymbol{f}_{m}}(\boldsymbol{x} + \boldsymbol{\delta}_{\boldsymbol{f}_{m}}^{*})] \in \mathbb{R}^{(K-1) \times m} .$$

Here, $\tilde{\boldsymbol{p}}_{\boldsymbol{f}}(\cdot)$ is an $(K-1)$-dimensional vector obtained by removing the $y$-th element of $\boldsymbol{p}_{\boldsymbol{f}}(\cdot)$ and normalizing with $l_{1}$ norm, and $\det(\mathbf{A}^{T}\mathbf{A})$ shows the square of volume of polytope spanned by $\tilde{\boldsymbol{p}}_{\boldsymbol{f}_{1}}(\boldsymbol{x} + \boldsymbol{\delta}_{\boldsymbol{f}_{1}}^{*}), \cdots, \tilde{\boldsymbol{p}}_{\boldsymbol{f}_{m}}(\boldsymbol{x} + \boldsymbol{\delta}_{\boldsymbol{f}_{m}}^{*})$ as in (Bernstein, 2009). Intuitively, the orthogonal probability vectors $\tilde{\boldsymbol{p}}_{\boldsymbol{f}_{1}}(\boldsymbol{x} + \boldsymbol{\delta}_{\boldsymbol{f}_{1}}^{*}), \cdots, \tilde{\boldsymbol{p}}_{\boldsymbol{f}_{m}}(\boldsymbol{x} + \boldsymbol{\delta}_{\boldsymbol{f}_{m}}^{*})$ could yield

**Algorithm 1** The AdvE$_{\text{OAP}}$ method

**Input:** training dataset $S$, number of base learners $m$, learning rate $\eta$ and the SGD iterations $T$
**Initialize:** base learners $\boldsymbol{f}_{1}, \cdots, \boldsymbol{f}_{m}$
**for** $t = 1$ **to** $T$ **do**
    Partition training dataset $S$ into batches $S_{1}, \cdots, S_{b}$
    **for** $k \in [b]$ **do**
        Generate adversarial examples for every base learner and example by the PGD-attack
        **for** $j = 1$ **to** $m$ **do**
            Update learner $\boldsymbol{f}_{j}$ by gradient descent in Eqn. (5)
        **end for**
    **end for**
**end for**
**Output:** Ensemble model $\bar{\boldsymbol{f}} = \sum_{j=1}^{m} \boldsymbol{f}_{j}/m$

larger volume and entropy in $\Gamma_{\alpha}(\boldsymbol{x}, y)$, and hence improve gradient diversity and cross diversity simultaneously.

Given training sample $S = \{(\boldsymbol{x}_{1}, y_{1}), \cdots, (\boldsymbol{x}_{n}, y_{n})\}$, we present the final objective optimization as

$$\sum_{i=1}^{n}\sum_{j=1}^{m} \max_{\boldsymbol{\delta} \in \Delta_{p}^{\epsilon}} \frac{\ell(f_{j}(\boldsymbol{x}_{i} + \boldsymbol{\delta}), y_{i}) - \lambda\Gamma_{\alpha}(\boldsymbol{x}_{i}, y_{i})}{mn} , \quad (5)$$

where $\lambda$ is a hyper-parameter to tradeoff adversarial loss and regularization $\Gamma_{\alpha}(\cdot)$. We could obtain mutual orthogonal base learners $\boldsymbol{f}_{1}, \cdots, \boldsymbol{f}_{m}$ from the optimization of Eqn. (5), and the details are given in Appendix D.2.

Figure 5 gives an illustration for the orthogonality of base learners. Here are three base learners in multi-class learning of 4 classes, and the ground-truth class is class 1. The output predictions are mutually orthogonal except for the ground-truth class 1. Therefore, the ensemble of three base learners could make correct prediction robustly even if three base learners are misled to different classes.

On the optimization of Eqn. (5), we take adversarial training method with stochastic gradient descent from (Madry et al., 2018). We generate adversarial examples w.r.t. all base learners by PGD-attack, and calculate gradients to update the parameters for each base learner of deep neural network. Algorithm 1 presents the detailed description for our Adversarial Ensemble training with Orthogonal Adversarial Predictions, which is short for AdvE$_{\text{OAP}}$. For Algorithm 1, the time complexity takes $m$-times as that of training a single neural network adversarially. In addition, it takes $O(m^{3})$ computational cost to calculate the regularization and gradients. More details are given in Appendix D.3

*Table 1.* Comparison of classification accuracies (mean $\pm$ std %) over adversarial examples generated by different adversarial attacks. •/∘ indicates that our AdvE$_{\text{OAP}}$ is significantly better/worse than the corresponding methods (pair-wise $t$-test at 95% significance level).

| | Methods | FGSM | PGD10 | PGD20 | PGD40 | AutoPGD | MORA | AutoAttack |
|---|---|---|---|---|---|---|---|---|
| **MNIST** | Our AdvE$_{\text{OAP}}$ | 96.413±0.124 | 95.676±0.039 | 95.648±0.056 | 95.504±0.098 | 94.907±0.187 | 94.672±0.209 | 94.072±0.423 |
| | GAL | 10.271±0.709• | 00.001±0.002• | 00.002±0.005• | 00.000±0.000• | 00.000±0.000• | 00.004±0.005• | 00.000±0.000• |
| | ADP | 10.888±1.931• | 00.000±0.000• | 00.000±0.000• | 00.000±0.000• | 00.000±0.000• | 00.004±0.006• | 00.000±0.000• |
| | AdvADP | 95.333±0.098• | 95.119±0.061 | 95.059±0.089 | 93.032±0.191• | 93.298±0.109• | 93.172±0.063• | 92.948±0.062• |
| | DVERGE | 74.903±1.059• | 37.506±4.292• | 34.807±4.981• | 05.846±2.573• | 00.001±0.002• | 00.299±0.361• | 00.000±0.000• |
| | PDD | 10.446±1.293• | 06.041±2.640• | 04.059±2.904• | 02.256±2.792• | 01.163±1.295• | 04.100±4.006• | 00.000±0.000• |
| | TRS | 91.044±0.892• | 86.544±1.300• | 86.709±1.014• | 85.954±3.283• | 80.902±4.905• | 79.628±5.471• | 78.615±5.997• |
| | iGAT$_{\text{ADP}}$ | 83.814±2.408• | 79.892±1.410• | 79.281±1.732• | 79.778±2.028• | 59.357±4.559• | 50.208±6.439• | 48.589±5.178• |
| **F-MNIST** | Our AdvE$_{\text{OAP}}$ | 82.743±0.263 | 81.770±0.245 | 81.752±0.248 | 80.543±0.290 | 81.467±0.079 | 80.643±0.332 | 80.506±0.327 |
| | GAL | 15.404±6.709• | 00.352±0.447• | 00.170±0.215• | 00.315±0.306• | 00.000±0.000• | 00.009±0.005• | 00.000±0.000• |
| | ADP | 22.287±1.741• | 00.001±0.002• | 00.001±0.002• | 00.000±0.000• | 00.000±0.000• | 00.009±0.005• | 00.000±0.000• |
| | AdvADP | 82.728±0.331 | 79.167±0.338• | 79.074±0.387• | 80.080±0.408• | 78.461±0.372• | 77.833±0.313• | 77.732±0.245• |
| | DVERGE | 48.242±1.536• | 27.140±2.878• | 25.846±3.019• | 32.652±3.033• | 17.222±4.786• | 20.126±4.605• | 15.035±5.560• |
| | PDD | 29.936±4.991• | 18.740±4.925• | 17.958±4.873• | 19.043±4.524• | 12.161±5.521• | 14.408±3.617• | 00.319±0.466• |
| | TRS | 70.767±0.466• | 69.330±0.125• | 69.285±0.105• | 67.641±0.440• | 68.719±0.098• | 67.800±0.183• | 67.250±0.304• |
| | iGAT$_{\text{ADP}}$ | 66.278±2.051• | 63.139±0.236• | 62.967±0.149• | 62.882±0.379• | 61.920±0.083• | 51.869±4.773• | 48.750±5.763• |
| **CIFAR10** | Our AdvE$_{\text{OAP}}$ | 55.718±0.245 | 53.076±0.249 | 52.996±0.255 | 52.903±0.295 | 51.997±0.234 | 48.318±0.065 | 47.884±0.060 |
| | GAL | 12.370±2.959• | 00.007±0.013• | 00.000±0.000• | 00.000±0.000• | 00.000±0.000• | 00.002±0.004• | 00.000±0.000• |
| | ADP | 23.100±0.757• | 00.008±0.010• | 00.001±0.003• | 00.000±0.000• | 00.000±0.000• | 00.001±0.003• | 00.000±0.000• |
| | AdvADP | 55.478±0.214 | 47.116±0.278• | 46.802±0.284• | 46.573±0.332• | 44.192±0.216• | 42.904±0.235• | 42.174±0.198• |
| | DVERGE | 28.536±0.882• | 05.358±0.344• | 04.830±0.343• | 04.690±0.269• | 02.246±0.155• | 02.868±0.178• | 01.748±0.164• |
| | PDD | 23.750±4.005• | 14.116±4.596• | 14.896±4.596• | 17.400±2.710• | 05.570±2.365• | 09.526±4.345• | 01.000±0.626• |
| | TRS | 39.350±0.402• | 37.548±0.431• | 37.453±0.440• | 37.263±0.259• | 36.690±0.440• | 32.828±0.433• | 32.588±0.433• |
| | iGAT$_{\text{ADP}}$ | 19.990±0.509• | 18.075±0.035• | 18.000±0.056• | 13.707±2.596• | 17.260±0.099• | 12.365±1.054• | 12.090±1.103• |

## 5. Experiments

We conduct experiments on three datasets[2]: MNIST of 70000 images and 784 dimensions, F-MNIST of 70000 images and 784 dimensions, and CIFAR10 of 60000 images and 3072 dimensions. Three datasets have been well-studied in previous works (Strauss et al., 2017; Kariyappa & Qureshi, 2019; Yang et al., 2021; Deng & Mu, 2024). We compare our method with the state-of-the-art methods on adversarial ensemble learning as follows:

- GAL: Non-adversarial training via the diversity of cos values of gradients (Kariyappa & Qureshi, 2019);

- ADP: Non-adversarial training via the diversity of the orthogonality of predictions (Pang et al., 2019);

- AdvADP: Adversarial training via the diversity of the orthogonality of predictions (Pang et al., 2019);

- DVERGE: Adversarial training on the exchange of adversarial examples in base learners to diversify the adversarial vulnerability (Yang et al., 2020);

---
[2]Download from https://paperswithcode.com/dataset.

- PDD: Non-adversarial training to diversify the feature representations via dropouts (Huang et al., 2021);

- TRS: GAL by preserving the smoothness of base learners (Yang et al., 2021);

- iGAT$_{\text{ADP}}$: AdvADP by allocating globally adversarial examples to base learners (Deng & Mu, 2024).

For all datasets, the perturbation size is set as 0.2, 0.05 and 0.03 under $l_{\infty}$-norm ball, respectively, as done in (Croce & Hein, 2020; Deng & Mu, 2024). For all methods, we select ResNet20 as base learners with learners number as 3, 3 and 8 for MNIST, F-MNIST and CIFAR10, respectively. More settings are given in Appendix E.1. All experiments are performed on a server with 64 CPU cores (2 Intel Xeon Gold 6430 CPUs) and NVIDIA GeForce RTX 4090 GPU, running Ubuntu 24.04 with 1TB main memory.

### 5.1. Performance under adversarial attacks

We take accuracy to measure performance on adversarial examples generated by seven popular adversarial attacks, i.e., FGSM (Goodfellow et al., 2015), PGD10, PGD20,

*Table 2.* Comparison of classification accuracies (mean ± std %) over adversarial examples generated by EOT and BPDA attacks. ●/○ indicates that our AdvE$_{OAP}$ is significantly better/worse than the corresponding methods (pair-wise $t$-test at 95% significance level).

| Attacks | Datasets | Our AdvE$_{OAP}$ | GAL | AdvADP | PDD | DVERGE | TRS | iGAT$_{ADP}$ |
|---|---|---|---|---|---|---|---|---|
| EOT | MNIST | 88.116±0.316 | 01.178±1.958● | 85.724±0.163● | 03.913±5.471● | 38.358±3.147● | 75.535±4.981● | 72.721±1.258● |
| | F-MNIST | 62.416±0.778 | 02.201±1.886● | 61.137±0.547● | 08.323±1.815● | 35.162±2.390● | 56.592±1.110● | 57.893±2.147● |
| | CIFAT10 | 42.003±0.504 | 01.490±0.179● | 40.787±0.238● | 08.535±0.006● | 20.371±0.550● | 31.629±1.156● | 16.408±2.697● |
| BPDA$_1$ | MNIST | 95.382±0.079 | 00.002±0.005● | 92.713±0.137● | 02.757±2.852● | 14.187±7.868● | 85.024±3.758● | 71.763±3.041● |
| | F-MNIST | 80.080±0.256 | 00.352±0.447● | 79.074±0.387● | 17.958±4.873● | 25.846±3.019● | 66.409±0.450● | 62.967±0.149● |
| | CIFAT10 | 49.530±0.050 | 00.000±0.000● | 44.687±0.286● | 14.896±4.596● | 04.730±0.343● | 33.417±0.310● | 10.693±1.776● |
| BPDA$_2$ | MNIST | 95.676±0.104 | 00.000±0.000● | 93.646±0.133● | 12.398±4.848● | 19.385±8.467● | 86.783±3.416● | 84.261±1.767● |
| | F-MNIST | 80.832±0.311 | 01.098±0.243● | 80.832±0.380 | 30.152±8.392● | 40.672±2.473● | 68.032±0.364● | 73.444±0.657● |
| | CIFAT10 | 53.092±0.230 | 00.000±0.000● | 47.858±0.174● | 25.277±4.449● | 05.608±0.343● | 37.383±0.355● | 14.500±2.608● |
| BPDA$_3$ | MNIST | 96.132±0.071 | 00.000±0.000● | 94.857±0.085● | 09.165±0.941● | 57.491±1.640● | 88.993±2.873● | 91.126±0.672● |
| | F-MNIST | 81.309±0.409 | 00.343±0.206● | 82.106±0.468 | 30.265±9.179● | 52.157±1.658● | 68.900±0.330● | 75.948±0.619● |
| | CIFAT10 | 52.440±0.226 | 00.000±0.000● | 46.240±0.252● | 25.690±3.737● | 05.347±0.382● | 36.480±0.261● | 13.600±2.548● |
| BPDA$_4$ | MNIST | 94.596±0.135 | 00.022±0.020● | 92.830±0.244● | 04.941±4.538● | 02.635±1.482● | 81.595±2.255● | 44.669±7.900● |
| | F-MNIST | 80.241±0.247 | 00.026±0.045● | 80.757±0.386 | 05.706±7.595● | 50.324±4.693● | 67.526±0.467● | 68.174±0.462● |
| | CIFAT10 | 53.383±0.110 | 00.273±0.073● | 53.397±0.160 | 12.650±1.642● | 26.573±0.046● | 40.507±0.399● | 14.410±1.161● |

*Table 3.* Comparison of accuracies (mean±std%) for our AdvE$_{OAP}$ with and without regularization $\Gamma(\cdot)$ under the APGD attack.

| Our AdvE$_{OAP}$ | MNIST | F-MNIST | CIFAR10 |
|---|---|---|---|
| without $\Gamma_\alpha(\cdot)$ | 93.513±0.046 | 80.317±0.159 | 50.833±0.102 |
| with $\Gamma_\alpha(\cdot)$ | 94.907±0.187 | 81.202±0.311 | 51.997±0.234 |
| Improvement | 1.394±0.149↑ | 1.150±0.140↑ | 1.163±0.257↑ |

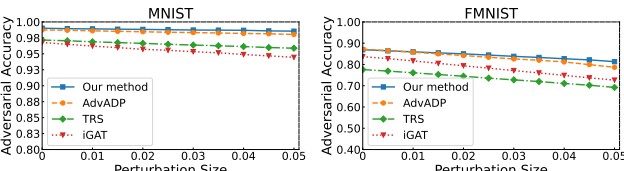

*Figure 6.* Influence of perturbation sizes under the PGD20 attack.

PGD40 (Madry et al., 2018), AutoPGD (Croce & Hein, 2020), MORA (Gao et al., 2022) and AutoAttack (Croce & Hein, 2020). All methods are evaluated over 50 runs with different random initializations, as summarized in Table 1.

From Table 1, it is clear that our AdvE$_{OAP}$ method achieves significantly better performance than GAL, PDD and TRS, since it wins at most times and never loses. This is because such methods merely focus on cos values as the diversity measure, yet ignore other factors such as cross diversity and individual adversarial losses, which is consistent with the ensemble decomposition as in Theorem 3.4.

Our AdvE$_{OAP}$ is also better than DEVERGE and iGAT$_{ADP}$, since DVERGE exchanges adversarial examples of base learners and iGAT$_{ADP}$ allocates adversarial examples of ensemble, without the consideration of diversities of base learners. Our AdvE$_{OAP}$ outperforms ADP and AdvADP, since such methods consider diversity over clean examples, rather than adversarial examples. It is quite different to study diversity in the adversarial ensemble learning, which is heavily relevant to intrinsic structures and output predictions of models simultaneously as in Theorem 3.1.

We further study two additional adversarial attacks BPDA (Athalye et al., 2018a) and EOT (Athalye et al., 2018b), where EOT considers adversarial perturbations insensitive to transformations, and BPDA considers potential gradient risks and designs corresponded attacks. More details are given in Appendix E.2. We implement EOT and four BPDA attacks, and experimental comparisons are summarized in Table 2. It is obvious that our AdvE$_{OAP}$ method achieves better performance than other adversarial ensembles, which shows the robustness of our method to gradient risks and adversarial perturbations insensitive to transformations.

We also present some ablation experiments to verify the effectiveness of the regularization $\Gamma_\alpha(\cdot)$ in Eqn. (5), which essentially considers gradient diversity and cross diversity via orthogonal adversarial predictions. Table 3 shows some experimental comparisons for our AdvE$_{OAP}$ with and without regularization. It is clear that our method takes better performance with regularization, which nicely shows the importance of diversity on the design of ensemble methods.

We study the influence of different perturbation size $\epsilon$ over datasets MNIST and FMNIST, as shown in Figure 6. It is observable that our AdvE$_{OAP}$ achieves better or comparable performance than other ensemble methods for different size of perturbations, in particular for larger $\epsilon$. This shows the effectiveness of our methods for larger perturbation size.

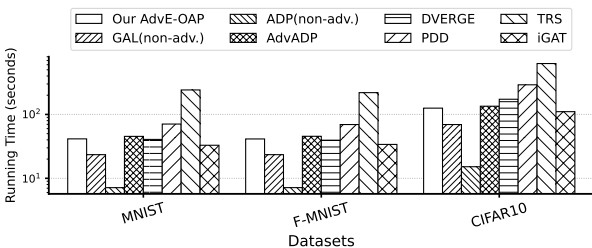

*Figure 7.* Comparisons of running time (seconds/epoch) for our AdvE$_{OAP}$ and other adversarial ensemble methods.

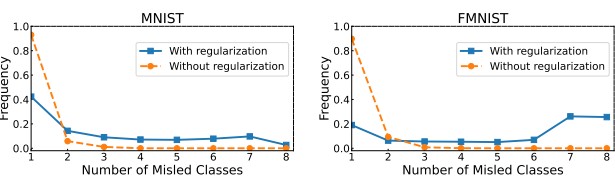

*Figure 8.* The frequency of number of misled classes.

We finally compare the training time of AdvE$_{OAP}$ with other methods in Figure 7. As can be seen, our AdvE$_{OAP}$ takes comparable training time to other adversarial-training ensembles AdvADP, DVERGE, PDD, TRS and iGAT, but with more time than the non-adversarial training methods GAL and ADP, which obviously take smaller adversarial prediction accuracy as shown in Table 1.

### 5.2. Orthogonality and convergence analysis

We now illustrate the orthogonality of base learners for our AdvE$_{OAP}$, which could mislead base learners to different classes under adversarial attacks. Figure 8 summarizes the frequency of number of misled classes with 8 base learners over two datasets MNIST and FMNIST. It is clear that base learners of our AdvE$_{OAP}$ predict with more different classes than that of AdvE$_{OAP}$ without regularization.

We also present the convergence analysis on adversarial ensemble loss, average of adversarial losses, as well as gradient diversity and cross diversity during the training process. Figure 9 presents the convergence curves over three datasets MNIST, FMNIST and CIFAR10. It is clear that our AdvE$_{OAP}$ method could decrease adversarial ensemble loss, and simultaneously increase gradient diversity and cross diversity. This is nicely in accordance with our diversity decomposition in Theorem 3.4.

## 6. Conclusion

Diversity has always been one of the most crucial factors on the designs of ensemble methods. This work focuses on the fundamental problems of diversity in the adversarial

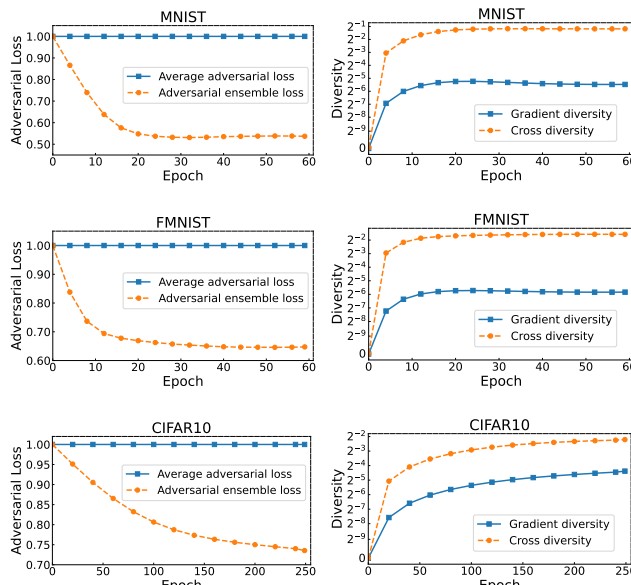

*Figure 9.* The curve of adversarial losses and diversities during the training process, where we normalize according to the average of individual adversarial losses.

ensemble learning. We prove the NP-Hard problem on the precise calculation of diversity for networks in adversarial ensemble learning, and give the first diversity decomposition under first-order approximation. Specifically, adversarial ensemble loss can be decomposed into average of individual adversarial losses, prediction diversity, gradient diversity and cross diversity. We consider similar diversity decomposition for classification with cross-entropy loss. Based on theoretical analysis, we develop a new ensemble method via orthogonal adversarial predictions to improve gradient and cross diversity simultaneously. An interesting future work is to explore other adversarial ensemble algorithms with better robustness and generalization from our theoretical analysis.

## Acknowledgements

The authors want to thank the reviewers for their helpful comments and suggestions. This research was supported by National Key R&D Program of China (2021ZD0112802) and NSFC (62376119).

## Impact Statement

This work shows the NP-Hardness of the calculation for diversity in adversarial ensemble learning and presents the first diversity decomposition with first-order approximation. It further develops a new ensemble method for adversarial defense and validates it through a series of experiments. There are many potential societal consequences of our work, none of which we feel must be specifically highlighted here.

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

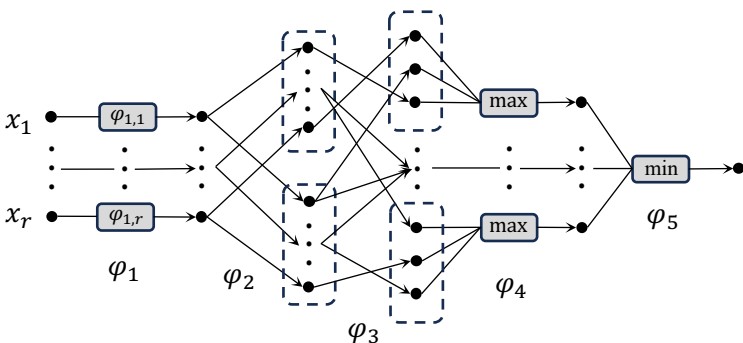

*Figure 10.* The structure of transformed neural network. The $\varphi_{1,j}(\cdot), \max(\cdot)$ and $\min(\cdot)$ functions can be constructed by ReLU functions. The $\varphi_2$ is fully connected layer with $-1, 1$ or $0$ weights. The $\varphi_3$ is fully connected layer with $1$ or $0$ weights.

## A. Proof of Theorem 3.1

**Definition A.1** (3-SAT problem)**.** Given $r$ boolean variables and $s$ clauses in a conjunctive normal form CNF formula with each clause's size at most 3, is there an assignment to the $r$ variables to make the CNF formula to be satisfied?

We present a key lemma for NP-hardness of adversarial squared loss, and the basic idea is a reduction from 3-SAT problem.

**Lemma A.2.** *For some ReLU neural network $g(\cdot)$ and example $(\boldsymbol{x}_0, y_0)$, it is an NP-hard problem to precisely calculate the following adversarial squared loss*

$$\max_{\boldsymbol{\delta} \in \Delta_p^\epsilon} \left\{ (g(\boldsymbol{x} + \boldsymbol{\delta}) - y)^2 \right\} .$$

*Proof.* It is sufficient to prove the NP-hardness of solving $\max_{\boldsymbol{\delta} \in \Delta_p^\epsilon}\{(g(\boldsymbol{x} + \boldsymbol{\delta}) - y)^2\} = \gamma$ for some $\gamma > 0$. Let $\phi = C_1 \wedge C_2 \wedge \cdots \wedge C_s$ be a 3-SAT formula over a variable set $V = \{v_1, \cdots, v_r\}$. Each $C_i = q_i^1 \vee q_i^2 \vee q_i^3$ is a disjunction with three literals $q_i^1, q_i^2, q_i^3$, and each literal is a variable from $V$ or their negations. The 3-SAT problem is to determine whether there exists an assignment $a : V \to \{0, 1\}$ for true $\phi$.

We will show that any 3-SAT formula $\phi$ can be transformed into a neural network $g$ over sample $(\boldsymbol{x}_0, y_0) = (\boldsymbol{0}, -1)$ in polynomial time, as well as the following sufficient and necessary condition:

$$\phi \text{ is satisfiable} \iff \max_{\boldsymbol{\delta} \in \Delta_p^\epsilon} \{(g(\boldsymbol{x}_0 + \boldsymbol{\delta}) - y_0)^2\} = (\epsilon/r^{1/p} + 1)^2 . \tag{6}$$

We will construct the ReLU neural network $g(\boldsymbol{x}) = \varphi_5 \circ \varphi_4 \circ \varphi_3 \circ \varphi_2 \circ \varphi_1(\boldsymbol{x})$ with input $\boldsymbol{x} = (x_1, x_2, \cdots, x_r) \in \mathbb{R}^r$ as shown in Figure 10. Specifically, we present the detailed constructions as follows:

- We construct function $\varphi_1(\boldsymbol{x}) : \mathbb{R}^r \to \mathbb{R}^r$ with the $j$-th element

$$[\varphi_1(\boldsymbol{x})]_j = \max(x_j + \epsilon/r^{1/p}, 0) - \max(x_j - \epsilon/r^{1/p}, 0) - \epsilon/r^{1/p} = \begin{cases} \epsilon/r^{1/p} & \text{for} \quad x_j > \epsilon/r^{1/p} \\ x_j & \text{for} \quad -\epsilon/r^{1/p} < x_j \leq \epsilon/r^{1/p} \\ -\epsilon/r^{1/p} & \text{for} \quad x_j \leq -\epsilon/r^{1/p} \end{cases} ,$$

  For $\boldsymbol{x}_0 = \boldsymbol{0}$ and $\|\boldsymbol{\delta}\|_p \leq \epsilon$, we can achieve the maximum or minimum of the elements of $\varphi_1(\boldsymbol{x}_0 + \boldsymbol{\delta})$ independently. It is easy to construct $\varphi_1(\boldsymbol{x})$ with a ReLU neural network because of three operators $+, -$ and $\max(\cdot, 0)$. Intuitively, the $i$-th element of $\varphi_1(\boldsymbol{x}_0 + \boldsymbol{\delta})$ can be viewed as the variable $v_i$ in 3-SAT problem, and its value $-\epsilon/r^{1/p}$ and $\epsilon/r^{1/p}$ can be viewed as the false and true of variable $v_i$, respectively.

- We construct $\varphi_2(\boldsymbol{x}) = (\boldsymbol{x}, -\boldsymbol{x})$, and this follows that, for $j \in [2r]$, $\boldsymbol{x}_0 = \boldsymbol{0}$ and $\|\boldsymbol{\delta}\|_p \leq \epsilon$,

$$-\epsilon/r^{1/p} \leq [\varphi_2 \circ \varphi_1(\boldsymbol{x}_0 + \boldsymbol{\delta})]_j \leq \epsilon/r^{1/p} .$$

  We achieve the maximum or minimum for the first $r$ elements of $\varphi_2 \circ \varphi_1(\boldsymbol{x}_0 + \boldsymbol{\delta})$ independently. The last $r$ elements are always the opposite of the first $r$ elements. Intuitively, the last $r$ elements of $\varphi_2 \circ \varphi_1(\boldsymbol{x}_0 + \boldsymbol{\delta})$ can be viewed as the negation of variables $v_1, \cdots, v_r$ in 3-SAT problem.

- We construct $\varphi_3(\boldsymbol{x}) = W\boldsymbol{x}$ with $W = (\boldsymbol{w}_1^1; \boldsymbol{w}_1^2; \boldsymbol{w}_1^3; \cdots; \boldsymbol{w}_s^1; \boldsymbol{w}_s^2; \boldsymbol{w}_s^3)^T \in \mathbb{R}^{3s \times 2r}$. Here, we construct three vectors $\boldsymbol{w}_i^1, \boldsymbol{w}_i^2, \boldsymbol{w}_i^3 \in \{0, 1\}^{2r}$ for clause $C_i = q_i^1 \vee q_i^2 \vee q_i^3$ for $i \in [s]$. For $k \in [3]$, the $\boldsymbol{w}_i^k$ is the unit vector with $j$-th and $(j+r)$-th element 1 if $q_i^k$ is variable $v_j$ and its negation $\neg v_j$, respectively. For $j \in [3s]$, $\boldsymbol{x}_0 = \boldsymbol{0}$ and $\|\boldsymbol{\delta}\|_p \leq \epsilon$, we have

$$-\epsilon/r^{1/p} \leq [\varphi_3 \circ \varphi_2 \circ \varphi_1(\boldsymbol{x}_0 + \boldsymbol{\delta})]_j \leq \epsilon/r^{1/p} .$$

  Intuitively, every three elements of $\varphi_3 \circ \varphi_2 \circ \varphi_1(\boldsymbol{x}_0 + \boldsymbol{\delta})$ can be viewed as three literals of a clause in 3-SAT problem. We achieve independently the maximum or minimum for $\varphi_3 \circ \varphi_2 \circ \varphi_1(\boldsymbol{x}_0 + \boldsymbol{\delta})$ whose corresponding literals are different variables.

- We construct $\varphi_4(\boldsymbol{x}) = (\max\{x_1, x_2, x_3\}, \max\{x_4, x_5, x_6\}, \cdots, \max\{x_{3s-2}, x_{3s-1}, x_{3s}\})$, where $\max\{\cdot, \cdot, \cdot\} = \max\{\max\{\cdot, \cdot\}, \cdot\}$ and $\min\{\cdot, \cdot, \cdot\} = \min\{\min\{\cdot, \cdot\}, \cdot\}$ can be constructed via ReLU function $(\max\{\cdot, 0\}$ function) as

$$\max\{a, b\} = \max\{a - b, 0\} + b \quad \text{and} \quad \min\{a, b\} = -\max\{b - a, 0\} + b . \tag{7}$$

  For $j \in [s]$, $\boldsymbol{x}_0 = \boldsymbol{0}$ and $\|\boldsymbol{\delta}\|_p \leq \epsilon$, we have

$$-\epsilon/r^{1/p} \leq [\varphi_4 \circ \varphi_3 \circ \varphi_2 \circ \varphi_1(\boldsymbol{x}_0 + \boldsymbol{\delta})]_j \leq \epsilon/r^{1/p} .$$

  We can achieve the maximum for the $j$-th element of $\varphi_4 \circ \varphi_3 \circ \varphi_2 \circ \varphi_1(\boldsymbol{x}_0 + \boldsymbol{\delta})$ if and only if the $(3j-2)$-th, $(3j-1)$-th or $3j$-th elements of $\varphi_3 \circ \varphi_2 \circ \varphi_1(\boldsymbol{x}_0 + \boldsymbol{\delta})$ are maximal. Intuitively, the $\max$ function can be viewed as the $\vee$ operator, and the $i$-th element of $\varphi_4 \circ \varphi_3 \circ \varphi_2 \circ \varphi_1(\boldsymbol{x}_0 + \boldsymbol{\delta})$ can be viewed as the clause $C_i$ in 3-SAT problem.

- We construct $\varphi_5(\boldsymbol{x}) = \min\{x_1, x_2, \cdots, x_s\} = \min\{\min\{\min\{\cdots, x_{s-2}\}, x_{s-1}\}, x_s\}$ via ReLU function. For $\boldsymbol{x}_0 = \boldsymbol{0}$ and $\|\boldsymbol{\delta}\|_p \leq \epsilon$, we have

$$-\epsilon/r^{1/p} \leq \varphi_5 \circ \varphi_4 \circ \varphi_3 \circ \varphi_2 \circ \varphi_1(\boldsymbol{x}_0 + \boldsymbol{\delta}) \leq \epsilon/r^{1/p} . \tag{8}$$

  We achieve the maximum if and only if all elements of $\varphi_4 \circ \varphi_3 \circ \varphi_2 \circ \varphi_1(\boldsymbol{x}_0 + \boldsymbol{\delta})$ are maximal. Intuitively, the $\min$ function can be viewed as the $\wedge$ operator in 3-SAT problem. The $-\epsilon/r^{1/p}$ and $\epsilon/r^{1/p}$ value of $\varphi_5 \circ \varphi_4 \circ \varphi_3 \circ \varphi_2 \circ \varphi_1(\boldsymbol{x}_0 + \boldsymbol{\delta})$ can be viewed as the false and true of the CNF formula $\phi = C_1 \wedge C_2 \wedge \cdots \wedge C_s$, respectively.

For $(\boldsymbol{x}_0, y_0) = (\boldsymbol{0}, -1)$ and $\epsilon \leq 1$, it remains to show that, from Eqns. (6) and (8),

$$\phi \text{ is satisfiable} \iff \text{there is an } \boldsymbol{\delta} \text{ s.t. } g(\boldsymbol{\delta}) = \epsilon/r^{1/p} . \tag{9}$$

$\implies$ If $\phi = C_1 \wedge C_2 \wedge \cdots \wedge C_m$ is satisfiable, then there is a satisfiable assignment $\alpha$. We set the $i$-th element of $\boldsymbol{\delta}$ as $\epsilon/r^{1/p}$ and $-\epsilon/r^{1/p}$ if $\alpha(v_i)$ is true and false, respectively. Then, we discuss $g(\boldsymbol{\delta})$ step by step as follows:

- The $i$-th element of $\varphi_1(\boldsymbol{\delta})$ is $-\epsilon/r^{1/p}$ and $\epsilon/r^{1/p}$ if the $i$-th element of $\boldsymbol{\delta}$ is $-\epsilon/r^{1/p}$ and $\epsilon/r^{1/p}$, respectively;

- The first $r$ elements of $\varphi_2 \circ \varphi_1(\boldsymbol{\delta})$ are equal to $\varphi_1(\boldsymbol{\delta})$ while the last $r$ elements are the opposite of the first $r$ elements;

- The $(3(i-1)+j)$-th element of $\varphi_3 \circ \varphi_2 \circ \varphi_1(\boldsymbol{\delta})$ is the $l$-th and $(l+r)$-th element of $\varphi_2 \circ \varphi_1(\boldsymbol{\delta})$ if literal $q_i^j$ is variable $v_l$ and $\neg v_l$, respectively;

- Every element of $\varphi_4 \circ \varphi_3 \circ \varphi_2 \circ \varphi_1(\boldsymbol{\delta})$ is $\epsilon/r^{1/p}$, since there is at least one true literal in $q_i^1, q_i^2, q_i^3$ for every satisfiable $C_i = q_i^1 \vee q_i^2 \vee q_i^3$, and hence there is at least one element with $\epsilon/r^{1/p}$ in every three elements of $\varphi_3 \circ \varphi_2 \circ \varphi_1(\boldsymbol{\delta})$;

- The final output $\varphi_5 \circ \varphi_4 \circ \varphi_3 \circ \varphi_2 \circ \varphi_1(\boldsymbol{\delta})$ is $\epsilon/r^{1/p}$, since $\varphi_5$ takes the minimum of $\varphi_4 \circ \varphi_3 \circ \varphi_2 \circ \varphi_1(\boldsymbol{\delta})$.

$\impliedby$ If $g(\boldsymbol{\delta}) = \epsilon/r^{1/p}$, we discuss $\boldsymbol{\delta}$ as follows:

- Every elements of $\varphi_4 \circ \varphi_3 \circ \varphi_2 \circ \varphi_1(\boldsymbol{\delta})$ is $\epsilon/r^{1/p}$ from $\varphi_5(\cdot) \in [-\epsilon/r^{1/p}, \epsilon/r^{1/p}]$;

- At least one element is equal to $\epsilon/r^{1/p}$ in every three elements of $\varphi_3 \circ \varphi_2 \circ \varphi_1(\boldsymbol{\delta})$, since $\varphi_4$ takes the maximum of every three elements. Without loss of generality, let all $(3(i-1)+1)$-th elements be $\epsilon/r^{1/p}$ for $i \in [m]$;

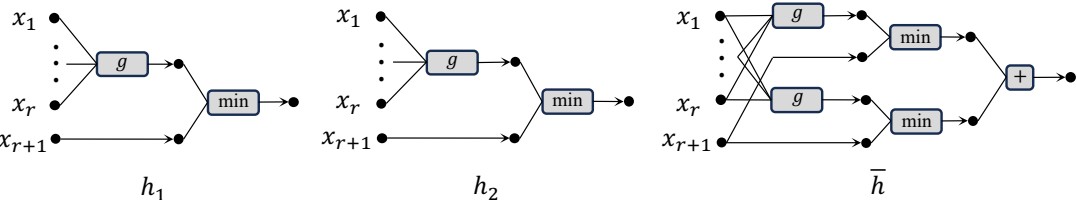

*Figure 11.* The structure of the transformed neural network. The $\varphi$ block is a network similarly in the proof of Theorem A.2.

- For the first $r$ elements of $\varphi_2 \circ \varphi_1(\boldsymbol{\delta})$, the $l$-th element will be equal to $\epsilon/r^{1/p}$ if literal $q_i^1$ is variable $v_l$. For the last $r$ elements of $\varphi_2 \circ \varphi_1(\boldsymbol{\delta})$, the $(l+r)$-th element will be equal to $\epsilon/r^{1/p}$ if literal $q_i^1$ is the negation of variable $\neg v_l$;

- The $l$-th element of $\varphi_1(\boldsymbol{\delta})$ is $\epsilon/r^{1/p}$ and $-\epsilon/r^{1/p}$ if the $l$-th element of $\varphi_2 \circ \varphi_1(\boldsymbol{\delta})$ is $\epsilon/r^{1/p}$ and $-\epsilon/r^{1/p}$, respectively;

- The $l$-th element of $\boldsymbol{\delta}$ is $\epsilon/r^{1/p}$ and $-\epsilon/r^{1/p}$ if the $l$-th element of $\varphi_1(\boldsymbol{\delta})$ is $\epsilon/r^{1/p}$ and $-\epsilon/r^{1/p}$, respectively.

Let $v_l$ be true and false if the $l$-th element of $\boldsymbol{\delta}$ is $\epsilon/r^{1/p}$ and $-\epsilon/r^{1/p}$, respectively. This assignment makes the CNF formula satisfiable, since there is at least one element in every three elements of $\varphi_3 \circ \varphi_2 \circ \varphi_1(\boldsymbol{\delta})$ with $\epsilon/r^{1/p}$. $\square$

*Proof of Theorem 3.1.* It is sufficient to prove the NP-hardness of solving

$$\frac{1}{2} \sum_{j=1}^2 \max_{\boldsymbol{\delta} \in \Delta_p^\epsilon} \{(f_j(\boldsymbol{x} + \boldsymbol{\delta}) - y)^2\} - \max_{\boldsymbol{\delta} \in \Delta_p^\epsilon} \{(\bar{f}(\boldsymbol{x} + \boldsymbol{\delta}) - y)^2\} = \gamma \quad \text{for some} \quad \gamma > 0 \,.$$

Let $\phi = C_1 \wedge C_2 \wedge \cdots \wedge C_s$ be a 3-SAT formula over variable set $V = \{v_1, \cdots, v_r\}$. Each $C_i = q_i^1 \vee q_i^2 \vee q_i^3$ is a disjunction with three literals $q_i^1, q_i^2, q_i^3$, and each literal is a variable from $V$ or their negations. We will show that any 3-SAT formula $\phi$ can be transformed into two neural networks $f_1$, $f_2$ and sample $(\boldsymbol{x}_0, y_0) = (\boldsymbol{0}, -1)$ in polynomial time, as well as the following sufficient and necessary condition:

$$\phi \text{ is satisfiable} \iff \Upsilon := \frac{1}{2} \sum_{j=1}^2 \max_{\boldsymbol{\delta} \in \Delta_p^\epsilon} \{(f_j(\boldsymbol{\delta}) + 1)^2\} - \max_{\boldsymbol{\delta} \in \Delta_p^\epsilon} \{(\bar{f}(\boldsymbol{\delta}) + 1)^2\} = (\epsilon/r^{1/p} + 1)^2 - 1 \,. \tag{10}$$

We will construct the ReLU neural network $f_1(\boldsymbol{x})$ and $f_2(\boldsymbol{x})$ with inputs $\boldsymbol{x} \in \mathbb{R}^{r+1}$ in Figure 11. For $\phi = C_1 \wedge C_2 \wedge \cdots \wedge C_s$, we construct neural network $g(\boldsymbol{x}_{1:r})$ as shown in Lemma A.2, where $\boldsymbol{x}_{1:r}$ are the first $r$ elements of $\boldsymbol{x}$. We construct $f_1(\boldsymbol{x}) = \min\{g(\boldsymbol{x}_{1:r}), \eta(x_{r+1})\}$ with

$$\eta(x_{r+1}) = \max(x_{r+1} + \frac{\epsilon}{r^{1/p}}, 0) - \max(x_{r+1} - \frac{\epsilon}{r^{1/p}}, 0) - \frac{\epsilon}{r^{1/p}} = \begin{cases} \epsilon/r^{1/p} & \text{for} \quad x_{r+1} > \epsilon/r^{1/p} \\ x_{r+1} & \text{for} \quad -\epsilon/r^{1/p} < x_{r+1} \leq \epsilon/r^{1/p} \\ -\epsilon/r^{1/p} & \text{for} \quad x_{r+1} \leq -\epsilon/r^{1/p} \,. \end{cases}$$

It is easy to construct $f_1(\boldsymbol{x})$ via a ReLU neural network from Eqn. (7), and we have

$$\phi \text{ is satisfiable} \iff \max_{\boldsymbol{\delta} \in \Delta_p^\epsilon} \{(f_1(\boldsymbol{\delta}) + 1)^2\} = (\epsilon/r^{1/p} + 1)^2 \,. \tag{11}$$

This is because $-\epsilon/r^{1/p} \leq f_1(\boldsymbol{x}) \leq \epsilon/r^{1/p}$ from Eqn. (8), and for $\epsilon \in (0, 1]$, $\phi$ is satisfiable if and only if there exists an $\boldsymbol{\delta}$ s.t. $f_1(\boldsymbol{\delta}) = \epsilon/r^{1/p}$. If $f_1(\boldsymbol{\delta}) = \epsilon/r^{1/p}$, then $\phi$ is satisfiable from Eqn. (9), since we have $g(\boldsymbol{\delta}_{1:r}) = \epsilon/r^{1/p}$ from $f_1(\boldsymbol{\delta}) = \min\{g(\boldsymbol{\delta}_{1:r}), \eta(\delta_{r+1})\}$ and $\eta(\delta_{r+1}) \leq \epsilon/r^{1/p}$. For the inverse direction, if $\phi$ is satisfiable, then there is an $\boldsymbol{\delta}_{1:r}$ s.t. $g(\boldsymbol{\delta}_{1:r}) = \epsilon/r^{1/p}$ from Eqn. (9). This follows that $f_1(\boldsymbol{\delta}) = \epsilon/r^{1/p}$ for $\delta_{k+1} = \epsilon/r^{1/p}$.

In a similar manner, we construct $f_2(\boldsymbol{x}) = \min\{g(\boldsymbol{x}_{1:k}), \eta(-x_{k+1})\}$, and have

$$\phi \text{ is satisfiable} \iff \max_{\boldsymbol{\delta} \in \Delta_p^\epsilon} \{(f_2(\boldsymbol{\delta}) + 1)^2\} = (\epsilon/r^{1/p} + 1)^2 \,. \tag{12}$$

For odd function $\eta(\boldsymbol{x})$, we have the ensemble $\bar{f}(\boldsymbol{x}) = (f_1(\boldsymbol{x}) + f_2(\boldsymbol{x}))/2$ as

$$\bar{f}(\boldsymbol{x}) = \frac{1}{2}(\min\{g(\boldsymbol{x}_{1:k}), \eta(x_{k+1})\} + \min\{g(\boldsymbol{x}_{1:k}), \eta(-x_{k+1})\}) \leq \frac{1}{2}(\eta(x_{k+1}) + \eta(-x_{k+1})) = \frac{1}{2}(\eta(x_{k+1}) - \eta(x_{k+1})) = 0.$$

We have $-\epsilon/r^{1/p} \leq \bar{f}(\boldsymbol{x}) \leq 0$, and it holds that $(\bar{f}(\boldsymbol{\delta}) + 1)^2 \leq 1$ for $\epsilon \in (0, 1]$, and the equality holds for $\boldsymbol{\delta} = \boldsymbol{0}$. We have

$$\max_{\|\boldsymbol{\delta}\|_p \leq \epsilon} \{(\bar{f}(\boldsymbol{\delta}) + 1)^2\} = 1. \tag{13}$$

For the proof of Eqn. (10), we have $\Upsilon = (\epsilon/r^{1/p} + 1)^2 - 1$ if $\phi$ is satisfiable from Eqns. (11)-(13). For the inverse direction, if $\Upsilon = (\epsilon/r^{1/p} + 1)^2 - 1$, then we have, from Eqn. (13),

$$\frac{1}{2} \sum_{j=1}^{2} \max_{\boldsymbol{\delta} \in \Delta_p^\epsilon} \{(f_j(\boldsymbol{\delta}) + 1)^2\} = (\epsilon/r^{1/p} + 1)^2. \tag{14}$$

From $f_1(\boldsymbol{\delta}), f_2(\boldsymbol{\delta}) \in [-\epsilon/r^{1/p}, \epsilon/r^{1/p}]$ and $\epsilon \in (0, 1]$, we have

$$(f_1(\boldsymbol{\delta}) + 1)^2 \leq (\epsilon/r^{1/p} + 1)^2 \quad \text{and} \quad (f_2(\boldsymbol{\delta}) + 1)^2 \leq (\epsilon/r^{1/p} + 1)^2.$$

This follows that, from Eqn. (14)

$$\max_{\boldsymbol{\delta} \in \Delta_p^\epsilon} \{(f_1(\boldsymbol{\delta}) + 1)^2\} = (\epsilon/r^{1/p} + 1)^2 \quad \text{and} \quad \max_{\boldsymbol{\delta} \in \Delta_p^\epsilon} \{(f_2(\boldsymbol{\delta}) + 1)^2\} = (\epsilon/r^{1/p} + 1)^2;$$

therefore, $\phi$ is satisfiable from Eqns. (11)-(12). This completes the proof. $\qquad\square$

## B. Appendix for Section 3.2

### B.1. Proof of Theorem 3.2

We begin with some useful lemmas as follows:

**Lemma B.1** (Hölder's inequality (Young, 1936)). *For two real vectors $\boldsymbol{a} = (a_1, \cdots, a_d)$ and $\boldsymbol{b} = (b_1, \cdots, b_d)$, we have*

$$\sum_{i=1}^{d} |a_i b_i| \leq \|\boldsymbol{a}\|_p \|\boldsymbol{b}\|_q \quad \text{for positive } p \text{ and } q \text{ with} \quad 1/p + 1/q = 1,$$

*where the equality holds if and only if $\alpha|a_i|^p = \beta|b_i|^q$ for $i \in [d]$ w.r.t. some positive constants $\alpha$ and $\beta$.*

**Lemma B.2.** *For vectors $\boldsymbol{w}, \boldsymbol{\delta} \in \mathbb{R}^d$, we have*

$$\max_{\|\boldsymbol{\delta}\|_p \leq \epsilon} \boldsymbol{w}^T \boldsymbol{\delta} = \epsilon\|\boldsymbol{w}\|_q \quad \text{and} \quad \min_{\|\boldsymbol{\delta}\|_p \leq \epsilon} \boldsymbol{w}^T \boldsymbol{\delta} = -\epsilon\|\boldsymbol{w}\|_q.$$

*Proof.* For every $\boldsymbol{\delta}$ with $\|\boldsymbol{\delta}\|_p \leq \epsilon$, we have, from Lemma B.1,

$$\boldsymbol{w}^T \boldsymbol{\delta} \leq \sum_{i=1}^{d} |w_i \delta_i| \leq \|\boldsymbol{\delta}\|_p \|\boldsymbol{w}\|_q \leq \epsilon\|\boldsymbol{w}\|_q.$$

Notice that the above equality holds if we choose $\boldsymbol{\delta} = \boldsymbol{\delta}^* = (\delta_1^*, \delta_2^*, \cdots, \delta_d^*)$ with

$$\delta_i^* = \text{sign}(w_i)\epsilon \left(|w_i|^q / \|\boldsymbol{w}\|_q^q\right)^{1/p},$$

where $\text{sign}(w_i)$ is equal to $-1, 0, 1$ if $w_i$ is negative, zero or positive, respectively. This follows that

$$\max_{\|\boldsymbol{\delta}\|_p \leq \epsilon} \boldsymbol{w}^T \boldsymbol{\delta} = \epsilon\|\boldsymbol{w}\|_q.$$

We also have, by letting $\boldsymbol{\delta}' = -\boldsymbol{\delta}$,

$$\min_{\|\boldsymbol{\delta}\|_p \leq \epsilon} \boldsymbol{w}^T \boldsymbol{\delta} = \min_{\|-\boldsymbol{\delta}'\|_p \leq \epsilon} \boldsymbol{w}^T(-\boldsymbol{\delta}') = \min_{\|\boldsymbol{\delta}'\|_p \leq \epsilon} -\boldsymbol{w}^T \boldsymbol{\delta}' = - \max_{\|\boldsymbol{\delta}'\|_p \leq \epsilon} \boldsymbol{w}^T \boldsymbol{\delta}' = -\epsilon\|\boldsymbol{w}\|_q,$$

which completes the proof. $\qquad\square$

**Lemma B.3.** *For linear function $h_{\boldsymbol{w}}(\boldsymbol{x}) = \boldsymbol{w}^T\boldsymbol{x} + b$, we have, for positive $p, q$ with $1/p + 1/q = 1$,*

$$\max_{\|\boldsymbol{\delta}\|_p \leq \epsilon} (\boldsymbol{w}^T(\boldsymbol{x}+\boldsymbol{\delta}) + b - y)^2 = (|\boldsymbol{w}^T\boldsymbol{x} + b - y| + \|\boldsymbol{w}\|_q\epsilon)^2 .$$

*Proof.* We have the adversarial squared loss

$$\max_{\|\boldsymbol{\delta}\|_p \leq \epsilon} (\boldsymbol{w}^T(\boldsymbol{x}+\boldsymbol{\delta}) + b - y)^2 = \max_{\|\boldsymbol{\delta}\|_p \leq \epsilon} (\boldsymbol{w}^T\boldsymbol{x} + b - y + \boldsymbol{w}^T\boldsymbol{\delta})^2 .$$

From Lemma B.2, we have $-\|\boldsymbol{w}\|_q\epsilon \leq \boldsymbol{w}^T\boldsymbol{\delta} \leq \|\boldsymbol{w}\|_q\epsilon$ and

$$\max_{\|\boldsymbol{\delta}\|_p \leq \epsilon} (\boldsymbol{w}^T(\boldsymbol{x}+\boldsymbol{\delta}) + b - y)^2 = \max_{\|\boldsymbol{\delta}\|_p \leq \epsilon} (\boldsymbol{w}^T\boldsymbol{x} + b - y + \boldsymbol{w}^T\boldsymbol{\delta})^2 = (|\boldsymbol{w}^T\boldsymbol{x} + b - y| + \|\boldsymbol{w}\|_q\epsilon)^2 ,$$

which completes the proof. $\qquad\square$

*Proof of Theorem 3.2.* We have the adversarial loss for $f_j$ and $\bar{f}$, from Lemma B.3,

$$\max_{\boldsymbol{\delta} \in \Delta_p^\epsilon}\{(\bar{f}(\boldsymbol{x}+\boldsymbol{\delta}) - y)^2\} = \max_{\boldsymbol{\delta} \in \Delta_p^\epsilon}\{(\bar{f}(\boldsymbol{x}) + \nabla\bar{f}(\boldsymbol{x})^T\boldsymbol{\delta} - y)^2\} = (|\bar{f}(\boldsymbol{x}) - y| + \|\nabla\bar{f}(\boldsymbol{x})\|_q\epsilon)^2 ,$$

$$\max_{\boldsymbol{\delta} \in \Delta_p^\epsilon}\{(f_j(\boldsymbol{x}+\boldsymbol{\delta}) - y)^2\} = \max_{\boldsymbol{\delta} \in \Delta_p^\epsilon}\{(f_j(\boldsymbol{x}) + \nabla f_j(\boldsymbol{x})^T\boldsymbol{\delta} - y)^2\} = (|f_j(\boldsymbol{x}) - y| + \|\nabla f_j(\boldsymbol{x})\|_q\epsilon)^2 .$$

This follows that

$$\frac{1}{m}\sum_{j=1}^m \max_{\boldsymbol{\delta} \in \Delta_p^\epsilon}\{(f_j(\boldsymbol{x}+\boldsymbol{\delta}) - y)^2\} - \max_{\boldsymbol{\delta} \in \Delta_p^\epsilon}\{(\bar{f}(\boldsymbol{x}+\boldsymbol{\delta}) - y)^2\}$$

$$= \frac{1}{m}\sum_{j=1}^m (|f_j(\boldsymbol{x}) - y| + \|\nabla f_j(\boldsymbol{x})\|_q\epsilon)^2 - (|\bar{f}(\boldsymbol{x}) - y| + \|\nabla\bar{f}(\boldsymbol{x})\|_q\epsilon)^2$$

$$\quad - \left((\bar{f}(\boldsymbol{x}) - y)^2 + 2|\bar{f}(\boldsymbol{x}) - y|\|\nabla\bar{f}(\boldsymbol{x})\|_q\epsilon + \|\nabla\bar{f}(\boldsymbol{x})\|_q^2\epsilon^2\right)$$

$$= \frac{\epsilon^2}{m}\sum_{j=1}^m (\|\nabla f_j(\boldsymbol{x})\|_q^2 - \|\nabla\bar{f}(\boldsymbol{x})\|_q^2) + \frac{2\epsilon}{m}\sum_{j=1}^m (\|\nabla f_j(\boldsymbol{x})\|_q|f_j(\boldsymbol{x}) - y| - \|\nabla\bar{f}(\boldsymbol{x})\|_q|\bar{f}(\boldsymbol{x}) - y|)$$

$$\quad + \frac{1}{m}\sum_{j=1}^m (f_j(\boldsymbol{x}) - y)^2 - (\bar{f}(\boldsymbol{x}) - y)^2$$

$$= \textbf{\textit{Gradient Diversity}} + \textbf{\textit{Cross Diversity}} + \frac{1}{m}\sum_{j=1}^m (f_j(\boldsymbol{x}) - y)^2 - (\bar{f}(\boldsymbol{x}) - y)^2 .$$

We also have

$$\frac{1}{m}\sum_{j=1}^m (f_j(\boldsymbol{x}) - y)^2 - (\bar{f}(\boldsymbol{x}) - y)^2 = \frac{1}{m}\sum_{j=1}^m f_j(\boldsymbol{x})^2 - \bar{f}(\boldsymbol{x})^2 = \frac{1}{m}\sum_{j=1}^m (f_j(\boldsymbol{x}) - \bar{f}(\boldsymbol{x}))^2 ,$$

which completes the proof. $\qquad\square$

## B.2. Training Details for Diversity Decomposition

We consider the base learners as convolutional neural network with two convolutional layers and one MLP layer of 100 neurons. The first convolutional layer has 24 filters with kernel size 5, while the second convolutional layer has 24 filters of kernel size 5. We take the ReLU activation function, and the input and output sizes are $3\times32\times32$ and 1, respectively.

We select the 5-th and 6-th class on datasets MNIST and F-MNIST to train the base learners independently, and take the SGD method (Robbins & Monro, 1951) with batch size 256 and learning rate 0.01. We consider the PGD-attack (Madry et al., 2018) to calculate the adversarial ensemble loss and average of individual adversarial losses. The perturbation size is set to 8/255 and 128/255 for $l_\infty$ and $l_2$ norm, respectively.

### B.3. Discussions on the Second-Order Approximation

For the second-order approximation, we have

$$f(\boldsymbol{x} + \boldsymbol{\delta}) \approx f(\boldsymbol{x}) + \nabla f(\boldsymbol{x})^T \boldsymbol{\delta} + \frac{1}{2} \boldsymbol{\delta}^T H(\boldsymbol{x}) \boldsymbol{\delta} \,,$$

where $H(\boldsymbol{x})$ is the Hessian matrix of $f$ at point $\boldsymbol{x}$. This follows that

$$\max_{\boldsymbol{\delta} \in \Delta_p^\epsilon} (f(\boldsymbol{x} + \boldsymbol{\delta}) - y)^2 = \max_{\boldsymbol{\delta} \in \Delta_p^\epsilon} \left( f(\boldsymbol{x}) + \nabla f(\boldsymbol{x})^T \boldsymbol{\delta} + \frac{1}{2} \boldsymbol{\delta}^T H(x) \boldsymbol{\delta} - y \right)^2 \,,$$

and hence, it is sufficient to consider two problems as follows

$$\max_{\boldsymbol{\delta} \in \Delta_p^\epsilon} f(\boldsymbol{x}) + \nabla f(\boldsymbol{x})^T \boldsymbol{\delta} + \frac{1}{2} \boldsymbol{\delta}^T H(x) \boldsymbol{\delta} - y \quad \text{and} \quad \min_{\boldsymbol{\delta} \in \Delta_p^\epsilon} f(\boldsymbol{x}) + \nabla f(\boldsymbol{x})^T \boldsymbol{\delta} + \frac{1}{2} \boldsymbol{\delta}^T H(x) \boldsymbol{\delta} - y \,.$$

We focus on the special case $p = 2$, and the above two problems can be formalized as

$$\min_{\|\boldsymbol{\delta}\|_2 \leq \epsilon} \frac{1}{2} \boldsymbol{\delta}^T B \boldsymbol{\delta} - \boldsymbol{b}^T \boldsymbol{\delta} \quad \text{for some } \boldsymbol{b} \in \mathbb{R}^n \text{ and symmetric matrix } B \in \mathbb{R}^{n \times n} \,. \tag{15}$$

This is known as the trust region subproblem (Forsythe & Golub, 1965; Moré & Sorensen, 1983; Fortin & Wolkowicz, 2004), and we have

- the optimal solution $\boldsymbol{\delta}^* = B^{-1}\boldsymbol{b}$ if there is no solution on the boundary $\{\boldsymbol{\delta} : \|\boldsymbol{\delta}\|_2 \leq \epsilon\}$ in (15), from the positive definiteness of $B$ and $\|B^{-1}\boldsymbol{b}\| < \epsilon$;

- the optimal solution $\boldsymbol{\delta}^* = -(B + \alpha^* I)^{-1}\boldsymbol{b}$ if there is solution on the boundary $\{\boldsymbol{\delta} : \|\boldsymbol{\delta}\|_2 \leq \epsilon\}$ in (15) and $\alpha^* > -\lambda_1$. Here, $\lambda_1$ is the smallest eigenvalue of $B$ and $\alpha^*$ is the solution of

$$\sum_{i=1}^{n} \frac{\gamma_j^2}{(\lambda_j + \alpha^*)^2} = \epsilon^2 \,, \tag{16}$$

  where $\lambda_1, \cdots, \lambda_n$ are eigenvalues of $B$ and $\gamma_1, \cdots, \gamma_n$ are the elements of $Q^T \boldsymbol{b}$ with eigendecomposition $B = Q \Lambda Q^T$.

- the optimal solution $\boldsymbol{\delta}^* = \boldsymbol{\delta}_0 + \tau \boldsymbol{z}$ if there is solution on the boundary $\{\boldsymbol{\delta} : \|\boldsymbol{\delta}\|_2 \leq \epsilon\}$ in (15) and $\alpha^* \leq -\lambda_1$ in (16). Here, $\boldsymbol{\delta}_0$ is the solution of
$$(B - \lambda_1 I)\boldsymbol{\delta}_0 = -\boldsymbol{b} \quad \text{s.t.} \quad \|\boldsymbol{\delta}_0\| \leq \epsilon \,,$$
  and $\boldsymbol{z}$ is an eigenvector of $B$ with eigenvalue $\lambda_1$ and $\tau \in \mathbb{R}$ satisfies $\|\boldsymbol{\delta}_0 + \tau \boldsymbol{z}\| = \epsilon$.

Here, the main challenge is how to obtain the closed-form solution of adversarial loss via second-order approximation, since it is relevant to the roots of high-order polynomials Eqn. (16).

### B.4. Discussions of Average of $\cos$ Values for $l_\infty$ Norm

We focus on 2-dimensional instance space $\mathcal{X} \subseteq \mathbb{R}^2$ and label space $\mathcal{Y} \subseteq \mathbb{R}$, and consider

$$f_1(\boldsymbol{x}) = x_1 + x_2, \quad f_2(\boldsymbol{x}) = x_1 - 3x_2, \quad f_3(\boldsymbol{x}) = abx_1 + bx_2 \quad \text{and} \quad f_4(\boldsymbol{x}) = abx_1 - bx_2 \,,$$

where $a = (\sqrt{5} - 1)/2$ and $b = \sqrt{2} + \sqrt{5}/5$. We study two ensembles: one ensemble of $f_1$ and $f_2$; the other ensemble of $f_3$ and $f_4$. For example $(\boldsymbol{x}, y) = ([1, 0], 1)$ and perturbation set $\Delta = \{\boldsymbol{\delta} : \|\boldsymbol{\delta}\|_\infty \leq 1\}$, we have

$$\cos(\nabla f_1(\boldsymbol{x}), \nabla f_2(\boldsymbol{x})) = \frac{\langle \nabla f_1(\boldsymbol{x}), \nabla f_2(\boldsymbol{x}) \rangle}{\|\nabla f_1(\boldsymbol{x})\|_2 \|\nabla f_2(\boldsymbol{x})\|_2} = \frac{\langle \nabla f_3(\boldsymbol{x}), \nabla f_4(\boldsymbol{x}) \rangle}{\|\nabla f_3(\boldsymbol{x})\|_2 \|\nabla f_4(\boldsymbol{x})\|_2} = \cos(\nabla f_3(\boldsymbol{x}), \nabla f_4(\boldsymbol{x})) \,,$$

and we also have the same average of individual adversarial losses from Lemma B.3 as follows

$$\sum_{i=1}^{2} \max_{\delta \in \Delta} \frac{(f_i(\boldsymbol{x} + \boldsymbol{\delta}) - y)^2}{2} = \sum_{i=3}^{4} \max_{\delta \in \Delta} \frac{(f_i(\boldsymbol{x} + \boldsymbol{\delta}) - y)^2}{2} \,.$$

However, the adversarial ensemble losses are different from

$$\max_{\delta \in \Delta}(f_1(\boldsymbol{x} + \boldsymbol{\delta})/2 + f_2(\boldsymbol{x} + \boldsymbol{\delta})/2 - y)^2 = 8 \quad \text{and} \quad \max_{\delta \in \Delta}(f_3(\boldsymbol{x} + \boldsymbol{\delta})/2 + f_4(\boldsymbol{x} + \boldsymbol{\delta})/2 - y)^2 \approx 7.2 \ .$$

Thus, there exist two ensembles of the same averages of cos values and individual adversarial losses, but with different adversarial ensemble losses for $l_\infty$ norm.

## C. Appendix for Section 3.3

### C.1. Proof of Lemma 3.3

We have the cross-entropy loss

$$\ell(f(\boldsymbol{x} + \boldsymbol{\delta}), y) = -y(f(\boldsymbol{x}) + \nabla f(\boldsymbol{x})^T \boldsymbol{\delta}) + \ln(1 + \exp(f(\boldsymbol{x}) + \nabla f(\boldsymbol{x})^T \boldsymbol{\delta})) \ ,$$

and the adversarial cross-entropy loss

$$\max_{\delta \in \Delta_p^\epsilon} \ell(f(\boldsymbol{x} + \boldsymbol{\delta}), y) = \max_{\delta \in \Delta_p^\epsilon} \left\{ -y(f(\boldsymbol{x}) + \nabla f(\boldsymbol{x})^T \boldsymbol{\delta}) + \ln(1 + \exp(f(\boldsymbol{x}) + \nabla f(\boldsymbol{x})^T \boldsymbol{\delta})) \right\} \ . \tag{17}$$

For $\boldsymbol{\delta} \in \Delta_p^\epsilon$, we have, from Lemma B.1

$$-\|\nabla f(\boldsymbol{x})\|_q \epsilon \le \nabla f(\boldsymbol{x})^T \boldsymbol{\delta} \le \|\nabla f(\boldsymbol{x})\|_q \epsilon \ .$$

We get the maximum of Eqn. (17) when

$$\nabla f(\boldsymbol{x})^T \boldsymbol{\delta} = \begin{cases} \|\nabla f(\boldsymbol{x})\|_q \epsilon & \text{for} \quad y = 0 \\ -\|\nabla f(\boldsymbol{x})\|_q \epsilon & \text{for} \quad y = 1 \ , \end{cases}$$

and the optimal adversarial perturbation $\boldsymbol{\delta}^*$ with $\nabla f(\boldsymbol{x})^T \boldsymbol{\delta}^* = -(2y - 1)\|\nabla f(\boldsymbol{x})\|_q \epsilon$. We finally have

$$f(\boldsymbol{x} + \boldsymbol{\delta}^*) = f(\boldsymbol{x}) + \nabla f(\boldsymbol{x})^T \boldsymbol{\delta}^* = f(\boldsymbol{x}) - (2y - 1)\|\nabla f(\boldsymbol{x})\|_q \epsilon \ ,$$

and the probability of the positive class of the adversarial example $\boldsymbol{x} + \boldsymbol{\delta}^*$

$$p_{f,+}^{adv} = \frac{1}{1 + \exp(-(f(\boldsymbol{x}) + \nabla f(\boldsymbol{x})^T \boldsymbol{\delta}^*))} = \frac{1}{1 + \exp(-(f(\boldsymbol{x}) - (2y - 1)\|\nabla f(\boldsymbol{x})\|_q \epsilon))} \ ,$$

which completes the proof. $\qquad\square$

### C.2. Proof of Theorem 3.4

We have

$$\ln(1 + \exp(f_j(\boldsymbol{x}) - y'\|\nabla f_j(\boldsymbol{x})\|_q \epsilon)) - \ln(1 + \exp(\bar{f}(\boldsymbol{x}) - y'\|\nabla \bar{f}(\boldsymbol{x})\|_q \epsilon))$$

$$= \frac{1}{1 + \exp(\bar{f}(\boldsymbol{x}) - y'\|\nabla \bar{f}(\boldsymbol{x})\|_q \epsilon)} \ln\left(\frac{1 + \exp(f_j(\boldsymbol{x}) - y'\|\nabla f_j(\boldsymbol{x})\|_q \epsilon)}{1 + \exp(\bar{f}(\boldsymbol{x}) - y'\|\nabla \bar{f}(\boldsymbol{x})\|_q \epsilon)}\right)$$

$$+ \frac{1}{1 + \exp(-(\bar{f}(\boldsymbol{x}) - y'\|\nabla \bar{f}(\boldsymbol{x})\|_q \epsilon))} \ln\left(\frac{1 + \exp(f_j(\boldsymbol{x}) - y'\|\nabla f_j(\boldsymbol{x})\|_q \epsilon)}{1 + \exp(\bar{f}(\boldsymbol{x}) - y'\|\nabla \bar{f}(\boldsymbol{x})\|_q \epsilon)}\right)$$

$$= \frac{1}{1 + \exp(\bar{f}(\boldsymbol{x}) - y'\|\nabla \bar{f}(\boldsymbol{x})\|_q \epsilon)} \ln\left(\frac{1 + \exp(f_j(\boldsymbol{x}) - y'\|\nabla f_j(\boldsymbol{x})\|_q \epsilon)}{1 + \exp(\bar{f}(\boldsymbol{x}) - y'\|\nabla \bar{f}(\boldsymbol{x})\|_q \epsilon)}\right)$$

$$+ \frac{1}{1 + \exp(-(\bar{f}(\boldsymbol{x}) - y'\|\nabla \bar{f}(\boldsymbol{x})\|_q \epsilon))} \ln\left(\frac{1 + \exp(-(f_j(\boldsymbol{x}) - y'\|\nabla f_j(\boldsymbol{x})\|_q \epsilon))}{1 + \exp(-(\bar{f}(\boldsymbol{x}) - y'\|\nabla \bar{f}(\boldsymbol{x})\|_q \epsilon))}\right)$$

$$+ \frac{1}{1 + \exp(-(\bar{f}(\boldsymbol{x}) - y'\|\nabla \bar{f}(\boldsymbol{x})\|_q \epsilon))}(f_j(\boldsymbol{x}) - y'\|\nabla f_j(\boldsymbol{x})\|_q \epsilon - \bar{f}(\boldsymbol{x}) + y'\|\nabla \bar{f}(\boldsymbol{x})\|_q \epsilon)$$

$$= KL(p_{\bar{f},adv}, p_{f_j,adv}) + p_{\bar{f},+,adv}(f_j(\boldsymbol{x}) - y'\|\nabla f_j(\boldsymbol{x})\|_q \epsilon - \bar{f}(\boldsymbol{x}) + y'\|\nabla \bar{f}(\boldsymbol{x})\|_q \epsilon) \ .$$

This follows that, from Lemma 3.3,

$$\frac{1}{m}\sum_{j=1}^{m}\max_{\|\boldsymbol{\delta}\|_p\leq\epsilon}\ell(\tilde{f}_j(\boldsymbol{x}+\boldsymbol{\delta}),y)-\max_{\|\boldsymbol{\delta}\|_p\leq\epsilon}\ell(\bar{g}(\boldsymbol{x}+\boldsymbol{\delta}),y)$$

$$=\frac{1}{m}\sum_{j=1}^{m}\ln(1+\exp(f_j(\boldsymbol{x})-y'\|\nabla f_j(\boldsymbol{x})\|_q\epsilon))-y(f_j(\boldsymbol{x})-y'\|\nabla f_j(\boldsymbol{x})\|_q\epsilon)$$

$$-\ln(1+\exp(\bar{f}(\boldsymbol{x})-y'\|\nabla\bar{f}(\boldsymbol{x})\|_q\epsilon))+y(\bar{f}(\boldsymbol{x})-y'\|\nabla\bar{f}(\boldsymbol{x})\|_q\epsilon)$$

$$=\frac{1}{m}\sum_{j=1}^{m}KL(p_{\bar{f},adv},p_{f_j,adv})+(p_{\bar{f},+,adv}-y)(2y-1)\epsilon\frac{1}{m}\sum_{j=1}^{m}(\|\nabla\bar{f}(\boldsymbol{x})\|_q-\|\nabla f_j(\boldsymbol{x})\|_q)\,,$$

which completes the proof. □

### C.3. Discussions of Average of $\cos$ Values for Cross-Entropy Loss

*Example* 2. There exist two ensembles of the same averages of $\cos$ values and individual adversarial cross-entropy losses, but with different adversarial ensemble losses for cross-entropy loss and $l_2$ norm.

*Proof.* We focus on 2-dimensional instance space $\mathcal{X}\subseteq\mathbb{R}^2$ and label space $\mathcal{Y}\subseteq\mathbb{R}$, and consider

$$f_1(\boldsymbol{x})=x_1+x_2,\quad f_2(\boldsymbol{x})=x_1-3x_2,\quad f_3(\boldsymbol{x})=abx_1+bx_2\quad\text{and}\quad f_4(\boldsymbol{x})=abx_1-bx_2\,,$$

where

$$a=\frac{\sqrt{5}-1}{2}\quad\text{and}\quad b=\frac{2\ln\left(\sqrt{(1+\exp(1+\sqrt{2}))(1+\exp(1+\sqrt{10}))}-1\right)}{\sqrt{5}-1+\sqrt{10-2\sqrt{5}}}\,.$$

We study two ensembles: one ensemble of $f_1$ and $f_2$; the other ensemble of $f_3$ and $f_4$. For example $(\boldsymbol{x},y)=([1,0],1)$ and perturbation set $\Delta=\{\boldsymbol{\delta}:\|\boldsymbol{\delta}\|_2\leq1\}$, we have

$$\cos(\nabla f_1(\boldsymbol{x}),\nabla f_2(\boldsymbol{x}))=\frac{\langle\nabla f_1(\boldsymbol{x}),\nabla f_2(\boldsymbol{x})\rangle}{\|\nabla f_1(\boldsymbol{x})\|_2\|\nabla f_2(\boldsymbol{x})\|_2}=\frac{\langle\nabla f_3(\boldsymbol{x}),\nabla f_4(\boldsymbol{x})\rangle}{\|\nabla f_3(\boldsymbol{x})\|_2\|\nabla f_4(\boldsymbol{x})\|_2}=\cos(\nabla f_3(\boldsymbol{x}),\nabla f_4(\boldsymbol{x}))\,,$$

and we also have the same average of individual adversarial losses from Lemma 3.3

$$\sum_{i=1}^{2}\max_{\delta\in\Delta}\frac{(f_i(\boldsymbol{x}+\boldsymbol{\delta})-y)^2}{2}=\sum_{i=3}^{4}\max_{\delta\in\Delta}\frac{(f_i(\boldsymbol{x}+\boldsymbol{\delta})-y)^2}{2}=\frac{\ln(1+\exp(1+\sqrt{2}))+\ln(1+\exp(1+\sqrt{10}))}{2}\,.$$

However, the adversarial ensemble losses are different from

$$\max_{\delta\in\Delta}(f_1(\boldsymbol{x}+\boldsymbol{\delta})/2+f_2(\boldsymbol{x}+\boldsymbol{\delta})/2-y)^2\approx2.4999\quad\text{and}\quad\max_{\delta\in\Delta}(f_3(\boldsymbol{x}+\boldsymbol{\delta})/2+f_4(\boldsymbol{x}+\boldsymbol{\delta})/2-y)^2\approx2.3738\,,$$

which completes the proof. □

*Example* 3. There exist two ensembles of the same averages of $\cos$ values and individual adversarial cross-entropy losses, but with different adversarial ensemble losses for cross-entropy loss for $l_\infty$ norm.

*Proof.* We focus on 2-dimensional instance space $\mathcal{X}\subseteq\mathbb{R}^2$ and label space $\mathcal{Y}\subseteq\mathbb{R}$, and consider

$$f_1(\boldsymbol{x})=x_1+x_2,\quad f_2(\boldsymbol{x})=x_1-3x_2,\quad f_3(\boldsymbol{x})=abx_1+bx_2\quad\text{and}\quad f_4(\boldsymbol{x})=abx_1-bx_2\,,$$

where $a=(\sqrt{5}-1)/2$ and $b=\ln(\sqrt{(1+\exp(3))(1+\exp(5))}-1)/\sqrt{5}$. We study two ensembles: one ensemble of $f_1$ and $f_2$; the other of $f_3$ and $f_4$. For example $(\boldsymbol{x},y)=([1,0],1)$ and perturbation set $\Delta=\{\boldsymbol{\delta}:\|\boldsymbol{\delta}\|_\infty\leq1\}$, we have

$$\cos(\nabla f_1(\boldsymbol{x}),\nabla f_2(\boldsymbol{x}))=\frac{\langle\nabla f_1(\boldsymbol{x}),\nabla f_2(\boldsymbol{x})\rangle}{\|\nabla f_1(\boldsymbol{x})\|_2\|\nabla f_2(\boldsymbol{x})\|_2}=\frac{\langle\nabla f_3(\boldsymbol{x}),\nabla f_4(\boldsymbol{x})\rangle}{\|\nabla f_3(\boldsymbol{x})\|_2\|\nabla f_4(\boldsymbol{x})\|_2}=\cos(\nabla f_3(\boldsymbol{x}),\nabla f_4(\boldsymbol{x}))\,,$$

and we also have the same average of individual adversarial losses from Lemma 3.3

$$\sum_{i=1}^{2} \max_{\delta \in \Delta} \frac{(f_i(\boldsymbol{x} + \boldsymbol{\delta}) - y)^2}{2} = \sum_{i=3}^{4} \max_{\delta \in \Delta} \frac{(f_i(\boldsymbol{x} + \boldsymbol{\delta}) - y)^2}{2} = \frac{\ln(1 + \exp(3)) + \ln(1 + \exp(5))}{2} .$$

However, the adversarial ensemble losses are different from

$$\max_{\delta \in \Delta}(f_1(\boldsymbol{x} + \boldsymbol{\delta})/2 + f_2(\boldsymbol{x} + \boldsymbol{\delta})/2 - y)^2 \approx 3.0486 \quad \text{and} \quad \max_{\delta \in \Delta}(f_3(\boldsymbol{x} + \boldsymbol{\delta})/2 + f_4(\boldsymbol{x} + \boldsymbol{\delta})/2 - y)^2 \approx 2.3738 ,$$

which completes the proof. $\square$

# D. Appendix for Section 4

### D.1. Proof of the Extended Diversity

For simplicity, we abbreviate $\boldsymbol{f}(\tilde{\boldsymbol{x}}_{\boldsymbol{f}})$ and $\boldsymbol{p}_{\boldsymbol{f}}(\tilde{\boldsymbol{x}}_{\boldsymbol{f}})$ to $\boldsymbol{f}$ and $\boldsymbol{p}_{\boldsymbol{f}}$, respectively. Let $f_k, p_{f_k}, k \in [K]$ be the $k$-th element of $\boldsymbol{f}$ and $\boldsymbol{p}_{\boldsymbol{f}}$, respectively. We have the adversarial cross-entropy loss for multi-classification

$$\ell(\boldsymbol{f}(\tilde{\boldsymbol{x}}_{\boldsymbol{f}}), y) = -f_y + \log\left(\sum_{k=1}^{K} \exp(f_k)\right) .$$

This follows that

$$\ell(\bar{\boldsymbol{f}}(\tilde{\boldsymbol{x}}_{\bar{\boldsymbol{f}}}), y) - \frac{1}{m}\sum_{j=1}^{m} \ell(\boldsymbol{f}_j(\tilde{\boldsymbol{x}}_{\boldsymbol{f}_j}), y) = -\bar{f}_y + \frac{1}{m}\sum_{j=1}^{m} f_{j,y} + \log\left(\sum_{k=1}^{K} \exp(\bar{f}_k)\right) - \frac{1}{m}\sum_{j=1}^{m} \log\left(\sum_{k=1}^{K} \exp(f_{j,k})\right) . \quad (18)$$

We also have

$$\log\left(\sum_{k=1}^{K} \exp(\bar{f}_k)\right) - \log\left(\sum_{k=1}^{K} \exp(f_{j,k})\right) = \sum_{k_1=1}^{K} \frac{\exp(\bar{f}_{k_1})}{\sum_{k=1}^{K} \exp(\bar{f}_k)} \log\left(\frac{\sum_{k=1}^{K} \exp(\bar{f}_k)}{\sum_{k=1}^{K} \exp(f_{j,k})}\right)$$

$$= \sum_{k_1=1}^{K} \frac{\exp(\bar{f}_{k_1})}{\sum_{k=1}^{K} \exp(\bar{f}_k)} \log\left(\frac{f_{j,k_1}/\sum_{k=1}^{K} \exp(f_{j,k})}{\bar{f}_{k_1}/\sum_{k=1}^{K} \exp(\bar{f}_k)} \times \frac{\bar{f}_{k_1}}{f_{j,k_1}}\right) = \sum_{k=1}^{K} p_{\bar{f}_k}(\bar{f}_k - f_{j,k}) - \mathrm{KL}(\boldsymbol{p}_{\bar{f}}, \boldsymbol{p}_{\boldsymbol{f}_j}) ,$$

and this follows that, by setting $\boldsymbol{r} = \boldsymbol{p}_{\bar{f}}(\tilde{\boldsymbol{x}}_{\bar{f}}) - \boldsymbol{e}_y$,

$$\frac{1}{m}\sum_{j=1}^{m} \log\left(\sum_{k=1}^{K} \exp(f_{j,k})\right) - \log\left(\sum_{k=1}^{K} \exp(\bar{f}_k)\right) = \sum_{j=1}^{m} \frac{\langle \boldsymbol{r}, \boldsymbol{f}_j(\tilde{\boldsymbol{x}}_{\boldsymbol{f}_j}) - \bar{\boldsymbol{f}}(\tilde{\boldsymbol{x}}_{\bar{f}})\rangle}{m} + \sum_{j=1}^{m} \frac{\mathrm{KL}(\boldsymbol{p}_{\bar{f}}(\tilde{\boldsymbol{x}}_{\bar{f}}), \boldsymbol{p}_{\boldsymbol{f}_j}(\tilde{\boldsymbol{x}}_{\boldsymbol{f}_j}))}{m} . \quad (19)$$

We have, from the first-order approximation $\boldsymbol{f}(\tilde{\boldsymbol{x}}_{\boldsymbol{f}}) \approx \boldsymbol{f}(\boldsymbol{x}) + \boldsymbol{J}_{\boldsymbol{f}}(\boldsymbol{x})\boldsymbol{\delta}_{\boldsymbol{f}}^*$,

$$\frac{1}{m}\sum_{j=1}^{m} \langle \boldsymbol{r}, \boldsymbol{f}_j(\tilde{\boldsymbol{x}}_{\boldsymbol{f}_j}) - \bar{\boldsymbol{f}}(\tilde{\boldsymbol{x}}_{\bar{f}})\rangle = \frac{1}{m}\sum_{j=1}^{m} \langle \boldsymbol{r}, \boldsymbol{J}_{\boldsymbol{f}_j}(\boldsymbol{x})\boldsymbol{\delta}_{\boldsymbol{f}_j}^* - \boldsymbol{J}_{\bar{f}}(\boldsymbol{x})\boldsymbol{\delta}_{\bar{f}}^*\rangle ,$$

which completes the proof by combining with Eqns. (18)-(19). $\square$

### D.2. Proof of Orthogonality

We now show the orthogonalization of the predictions of base learners by optimizing Eqn. (5) as follows.

**Theorem D.1.** *For $K$ multi-class learning, let $f_1, \cdots, f_m$ be $m$ base learners with cross-entropy loss being at least $B$, and $K - 1$ is a multiple of $m$. We have the minimizer of Eqn. (5) over example $(\boldsymbol{x}, y)$ as*

$$p_{f_i}(\boldsymbol{x} + \boldsymbol{\delta}_{f_i}^*)_k = \begin{cases} \exp(-B) & for \quad k = y \\ \frac{1 - \exp(-B)}{K-1} & for \quad k \in s_i \\ 0 & otherwise , \end{cases} \quad (20)$$

*where $s_1, \cdots, s_m$ is a partition of set $\{1, \cdots, K\}\setminus\{y\}$ and $p_{f_i}(\tilde{\boldsymbol{x}}_{f_i})_k$ is the $k$-th element of $p_{f_i}(\tilde{\boldsymbol{x}}_{f_i})$.*

*Table 4.* Comparison of training time (times(s) per epoch) for our AdvE$_{\text{OAP}}$ with and without the regularization $\Gamma$.

| Our AdvE$_{\text{OAP}}$ | MNIST | F-MNIST | CIFAR10 |
|---|---|---|---|
| without regularization $\Gamma_\alpha(\cdot)$ | 40.08 | 40.11 | 123.84 |
| with regularization $\Gamma_\alpha(\cdot)$ | 41.45 | 41.32 | 125.48 |

*Proof.* We first have $p_{f_i}(\boldsymbol{x} + \boldsymbol{\delta}^*_{f_i})_y = \exp(-B)$, since the adversarial cross-entropy loss is at least $B$ for each base learner. The regularization in Eqn. (5) is defined as

$$\Gamma_\alpha(\boldsymbol{x}, y) = H\left(\sum_{j=1}^{m} \tilde{\boldsymbol{p}}_{\boldsymbol{f}_j}(\tilde{\boldsymbol{x}}_{\boldsymbol{f}_j})/m\right) + \alpha \cdot \log(V(\tilde{\boldsymbol{p}}_{\boldsymbol{f}_1}(\tilde{\boldsymbol{x}}_{\boldsymbol{f}_1}), \cdots, \tilde{\boldsymbol{p}}_{\boldsymbol{f}_m}(\tilde{\boldsymbol{x}}_{\boldsymbol{f}_m}))) \ .$$

The $V(\tilde{\boldsymbol{p}}_{\boldsymbol{f}_1}(\tilde{\boldsymbol{x}}_{\boldsymbol{f}_1}), \cdots, \tilde{\boldsymbol{p}}_{\boldsymbol{f}_m}(\tilde{\boldsymbol{x}}_{\boldsymbol{f}_m}))$ achieves its maximum if and only if the non-label probability vectors of each individual network are mutually orthogonal (Bernstein, 2009). The $H(\sum_{j=1}^{m} \tilde{\boldsymbol{p}}_{\boldsymbol{f}_j}(\tilde{\boldsymbol{x}}_{\boldsymbol{f}_j})/m)$ achieves the maximum if and only if the mean of non-label probability vectors of individual networks are uniform. It is obvious that Eqn. (20) satisfies the two conditions simultaneously. Thus, Eqn. (20) is the minimizer of Eqn. (5). $\qquad\square$

### D.3. Other Details of Algorithm 1

**PGD-attack for $l_p$ norm perturbation ball**

The PGD-attack generates adversarial examples iteratively for $l_\infty$-norm perturbation ball as follows

$$\boldsymbol{x}^{t+1,j} = \prod_{\boldsymbol{x}+\Delta_\infty^\epsilon} (\boldsymbol{x}^{t,j} + \alpha \cdot \text{sign}(\nabla\ell(f_j(\boldsymbol{x}^{t,j}), y))) \ ,$$

where $\prod$ denotes the projection and $\boldsymbol{x} + \Delta_\infty^\epsilon = \{\boldsymbol{x} + \boldsymbol{\delta} | \boldsymbol{\delta} \in \Delta_\infty^\epsilon\}$. The $\boldsymbol{x}^{0,j}$ is initialized as $\boldsymbol{x}$, and $\boldsymbol{x}^{T,j}$ is used as the adversarial example of base learner $f_j$. For other $l_p$ norm perturbation ball, PGD-attack generates adversarial examples as

$$\boldsymbol{x}^{t+1,j} = \prod_{\boldsymbol{x}+\Delta_p^\epsilon} (\boldsymbol{x}^{t,j} + \alpha \cdot g_p(\nabla\ell(f_j(\boldsymbol{x}^{t,j}), y))) \ ,$$

where $g_p$ is the function that maps the gradient to the update direction

$$g_p(\boldsymbol{w}) = \text{sign}(w_i)\epsilon(|w_i|^q/\|\boldsymbol{w}\|_q^q)^{1/p} \ .$$

**Calculating the volume of polytope**

For $m$ vectors $\boldsymbol{x}_1, \cdots, \boldsymbol{x}_m \in \mathbb{R}^d$ and $X = (\boldsymbol{x}_1, \cdots, \boldsymbol{x}_m) \in \mathbb{R}^{d \times m}$, we have, from matrix theory (Bernstein, 2009),

$$V^2(\boldsymbol{x}_1, \cdots, \boldsymbol{x}_m) = \det(X^T X) \ ,$$

where $\det(X^T X)$ is the determination of the matrix $X^T X$. For matrix $A \in \mathbb{R}^{n \times n}$, we also have

$$\frac{\partial \det(A)}{\partial a_{ij}} = \det(A)(A^{-1})^T \ ,$$

where $a_{ij}$ is the $i$-th row and $j$-th column element of $A$. We finally optimize the objective Eqn. (5) for neural networks with SGD method (Robbins & Monro, 1951).

**Time Complexity of Algorithm 1**

The time complexity of Algorithm 1 takes $m$-times as that of training a single neural network adversarially ($m$ is the number of neural networks in the ensemble). In addition, it takes $O(m^3)$ computational cost for the regularization with its gradient. In practice, the regularization takes much smaller computational cost than that of training neural networks, as in Table 4.

*Table 5.* Hyperparameters of all ensemble methods used in our experiments. Parameters that were not applicable were left blank.

| Parameter | GAL | ADP | AdvADP | DVERGE | PDD | TRS | iGAT(ADP) |
|-----------|-----|-----|--------|--------|-----|-----|-----------|
| $\alpha$ | 0.5 | 2 | 2 | - | 0.01 | 1 | 2 |
| $\beta$ | - | 0.5 | 0.5 | - | - | 5 | 0.5 |

# E. Appendix for Section 5

### E.1. Experimental settings

For iGAT$_{\text{ADP}}$, we take 150, 150 and 480 epoches for MNIST, F-MNIST and CIFAR10 for convergence; while for other ensemble methods, we take 60, 60 and 250 epoches for MNIST, F-MNIST and CIFAR10, respectively. For adversarial examples in training process, we take PGD10 with 10 steps with step-size 0.04, 0.01 and 0.008 for MNIST, F-MNIST and CIFAR10, respectively. We set $\alpha = 0.02$ and $\lambda = 10$ for our method, and Table 5 summarizes parameter setting for others.

### E.2. EOT and BPDA attacks

We take the Backward Pass Differentiable Approximation (BPDA) attack (Athalye et al., 2018a) for potential gradient risks and designs attacks. We design four different attacks as follows:

- BPDA$_1$: For potential gradients vanishing risk (i.e., small gradient of ensemble from different gradient of base learner), we instead use the $k$ times of the average logits of the base learners as the logit of the ensemble, where $k \in [1, m]$ and $m$ is the number of base learners. We evaluate all possible values of $k \in [1, m]$ and report the lowest adversarial accuracy observed.

- BPDA$_2$: For potential incorrect gradients from random gradients of single base learner, we consider the attack of deleting a base learner and using other gradients of the ensemble.

- BPDA$_3$: For potential incorrect gradients from random gradients of base learners, we could consider the attack of selecting randomly half of base learners at each step for attack.

- BPDA$_4$: For other potential gradient risks, we consider the black box attack (Andriushchenko et al., 2020).

We also take the Expectation over Transformation (EOT) attack (Athalye et al., 2018b) to add adversarial perturbations insensitively in transformations. We implement 20 times rotations randomly within -30 to +30 degrees.

