# OpenReview forum: "On the Diversity of Adversarial Ensemble Learning"
_ICML.cc/2025/Conference — ICML 2025 poster_

### Official Review · Reviewer_ybcr · 2025-02-19

**Overall Recommendation:** 3

**Summary:**

This paper investigates the role of diversity in adversarial ensemble learning, addressing two key questions: how to define diversity formally in adversarial scenario and how diversity correlates. To address this questions, the paper introduces a first-order approximation and proposes a novel diversity decomposition within four components: average of individual adversarial losses, prediction diversity, gradient diversity, and cross diversity. Empirical evaluations validate the effectiveness of proposed AdvEOAP method.

## Update After Rebuttal
Dear Author,

Thank you for your detailed response and clarification. My concerns have mostly been addressed, and I will raise my score.

Please remember to organize the rebuttal materials mentioned above in your next revision, as this will greatly enhance the clarity of the paper.

Best regards,\
Reviewer ybcr

**Claims And Evidence:**

The major claims are well supported by empirical or theoretical evidences.

**Essential References Not Discussed:**

Not applicable.

**Experimental Designs Or Analyses:**

The experimental designs make sense to me.

**Methods And Evaluation Criteria:**

The methods and evaluation criteria make sense to me.

**Other Comments Or Suggestions:**

None

**Other Strengths And Weaknesses:**

Strengths:

1. The NP-Hardness proof is well-motivated and distinguishes adversarial ensemble learning from traditional ensemble theory.
2. The paper is technically sound, with rigorous mathematical formulations and proofs provided for key theorems that underpin the main claims.
3. This paper is well-written, making it easy to follow.

Weaknesses:

1. It would benefit to examine the performance of the authors’ proposed method with Expectation over Transformation (EOT) and Backward Pass Differentiable Approximation (BPDA).
2. The practical applicability of the method in real-world scenarios is not fully explored, such as large-scale Imagenet-1K and Vit architecture.
3. Understanding the computational overhead introduced by the proposed method is essential, especially when compared to other baseline methods.

**Questions For Authors:**

Refer to strengths and weaknesses.

**Relation To Broader Scientific Literature:**

Not applicable.

**Theoretical Claims:**

I only reviewed the general formula derivation without verifying the detailed proofs.

---

> ### Author Rebuttal · Authors · 2025-04-01
>
> [Q1] It would benefit to examine the performance of the authors’ proposed method with Expectation over Transformation (EOT) and Backward Pass Differentiable Approximation (BPDA).
>
> [A1] We will clarify that this work does not consider the EOT and BPDA attacks, because the EOT attack is designed for random networks [Athalye et al., 2017], while the BPDA attack is considered for networks with non-differentiable operations [Athalye et al., 2018]. In contrast, our model is deterministic without any non-differentiable operations.
>
> [Q2] The practical applicability of the method in real-world scenarios is not fully explored, such as large-scale Imagenet-1K and Vit architecture.
>
> [A2] We will clarify that we consider ResNet20 on datasets Minist, F-minist and Cifar10 for fair comparisons with previous ensemble methods, and we have been conducting experiments on larger datasets with complex network architectures. Here are some preliminary results:
>
> Adversarial accuracy for CIFAR100 with 5 ResNet18
>
> | Methods | PGD20 | AutoAttack |
> |-|-|-|
> | Our Method | 25.16 | 24.39 |
> | AdvADP | 23.690 | 21.22 |
>
> We could present more experimental results in the following days due to computational resources. For example, it is about 4 days to train a deep neural network on Tiny-ImageNet with four A6000 GPUs, while we only have a single NVIDIA GeForce RTX 4090 GPU to train multiple deep neural networks for diversity.
>
> [Q3] Understanding the computational overhead introduced by the proposed method is essential, especially when compared to other baseline methods.
>
> [A3] We will clarify that, for AdvEOAP, the training time takes $m$-times as that of training a single neural network ($m$ is the number of neural networks in the ensemble). In addition, it takes $O(m^3)$ computational cost for the regularization with its gradients.

---

> > ### Comment · Reviewer_ybcr · 2025-04-01
> >
> > Dear Author,
> >
> > Thank you for your response.
> >
> > Regarding Q1, the EOT and BPDA techniques are designed to handle the potential pseudo-robustness in more challenging scenarios. Therefore, they are definitely applicable to the proposed AdvEOAP which is differentiable.
> >
> > For Q2, considering the limited rebuttal time, my concern has been partially addressed.
> >
> > Regarding Q3, considering the $O(m^3)$) computational cost associated with the regularization, I am concerned about the fairness and scalability of the proposed method. For a fair comparison, what would the performance look like if AdvEOAP and baseline methods (e.g., AdvADP) are trained for exactly the same GPU time? Regarding scalability, if we require to ensemble bench of models, such as more than 20 single neural networks, is the proposed method still practicable in an affordable time?
> >
> > Since the authors did not address my major concerns well, I will lower my score.
> >
> > Best regards,\
> > Reviewer ybcr

---

> > > ### Author Response · Authors · 2025-04-03
> > >
> > > [Q1] Regarding Q1, the EOT and BPDA techniques … definitely applicable to the proposed AdvEOAP...
> > >
> > > [A1] We will clarify that EOT is applicable to the AdvEOAP by adding adversarial perturbations insensitively to transformations as in [Athalye et al. ICML2018a], and we implement 20 times rotations randomly within -30 to +30 degrees. Empirical results of adversarial accuracy (%) are shown as follows:
> > > | Datasets | Our method | AdvADP | TRS |
> > > |-|-|-|-|
> > > | MNIST | 88.41 | 85.84 | 77.12 |
> > > | F-MNIST | 62.43 | 61.75 | 55.06 |
> > > | CIFAR10 | 42.51 | 41.21 | 31.22 |
> > >
> > > We will also clarify that BPDA is applicable to the AdvEOAP by considering potential risks and designing attacks as in [Athalye et al. ICML2018b], and we will add empirical results as follows:
> > >
> > > i) For potential gradients vanishing risk (i.e., small gradient of ensemble from different gradient of base learner), we instead consider the attack of $m$ times of gradients rather than the original gradients, and empirical results of adversarial accuracy (%) are shown as follows:
> > >
> > > | Datasets | Our method | AdvADP | TRS |
> > > |-|-|-|-|
> > > | MNIST | 95.40 | 92.70 | 85.86 |
> > > | F-MNIST | 80.32 | 79.07 | 66.35 |
> > > | CIFAR10 | 49.68 | 46.82 | 33.05 |
> > >
> > > We could also consider the attack of selecting randomly a single base learner at each step, and empirical results of adversarial accuracy (%) are shown as follows:
> > >
> > > | Datasets | Our method | AdvADP | TRS |
> > > |-|-|-|-|
> > > | MNIST | 96.21 | 94.98 | 89.41 |
> > > | F-MNIST | 81.68 | 81.60 | 68.82 |
> > > | CIFAR10 | 55.77 | 51.97 | 38.93 |
> > >
> > > ii) For potential incorrect gradients from random gradients of base learners, we consider the attack of deleting a single base learner and using other gradients of the ensemble, and empirical results of adversarial accuracy (%) are shown as follows:
> > >
> > > | Datasets | Our method | AdvADP | TRS |
> > > |-|-|-|-|
> > > | MNIST | 95.79 | 94.03 | 87.58 |
> > > | F-MNIST | 81.30 | 80.67 | 68.08 |
> > > | CIFAR10 | 53.46 | 47.84 | 37.05 |
> > >
> > > iii) For other potential risks, we could consider the attack of using the adversarial training [Madry et al. 2018] model as a surrogated model, and empirical results of adversarial accuracy (%) are shown as follows:
> > > | Datasets | Our method | AdvADP | TRS |
> > > |-|-|-|-|
> > > | MNIST | 96.12 | 95.36 | 93.42 |
> > > | F-MNIST | 82.79 | 82.40 | 73.13 |
> > > | CIFAR10 | 53.58 | 50.41 | 47.90 |
> > >
> > > We could also consider the black box attack [Andriushchenko et al. 2020], and empirical results of adversarial accuracy (%) are shown as follows:
> > > | Datasets | Our method | AdvADP | TRS |
> > > |-|-|-|-|
> > > | MNIST | 94.49 | 92.71 | 83.85 |
> > > | F-MNIST | 80.33 | 79.92 | 66.84 |
> > > | CIFAR10 | 53.72 | 53.17 | 40.22 |
> > >
> > > Our AdvEOAP always achieves better performance than other adversarial ensemble methods under EOT and BPDA attacks.
> > >
> > >
> > > [Q2] …the $O(m^3)$ computational cost associated with the regularization…the fairness and scalability of the proposed method…what would the performance look like if AdvEOAP and baseline methods (e.g., AdvADP) are trained for exactly the same GPU time…is the proposed method still practicable in an affordable time?
> > >
> > > [A2] We will clarify that
> > >
> > > $$\text{total computational cost} = \text{computational cost for neural networks} + O(m^3)  \text{ computational cost for regularization}$$
> > >
> > > where the regularization takes much smaller computational cost than that of training neural networks. We will add the running-time (time(s) per epoch) comparisons as follows:
> > >
> > > | Datasets | without regularization | with regularization |
> > > |-|-|-|
> > > | MNIST | 40.08 | 41.45 |
> > > | F-MNIST | 40.11 | 41.32 |
> > > | CIFAR10 | 123.84 | 125.48 |
> > >
> > >
> > > We will also clarify that our method takes comparable running time to ensemble methods with adversarial training [Madry et al. 2018], but more running time than other ensemble methods without adversarial training, which takes smaller adversarial prediction accuracy obviously. We will add the running time (time(s) per epoch) comparisons as follows.
> > >
> > > | Datasets | GAL (non-adv.) | ADP (non-adv.) | AdvADP | DVERGE | PDD | TRS | $\text{iGAT}_{\text{ADP}}$ | Our $\text{AdvE}_{\text{OAP}}$ |
> > > |-|-|-|-|-|-|-|-|-|
> > > | MNIST | 23.45 | 7.14 | 45.36 | 40.65 | 70.72 | 241.8 | 32.93 | 41.45 |
> > > | F-MNIST | 23.45 | 7.15 | 45.50 | 39.52 | 69.48 | 219.32 | 33.94 | 41.32 |
> > > | CIFAR10 | 69.47 | 15.18 | 134.54 | 172.93 | 291.07 | 626.33 | 110.51 | 125.48 |

---

### Official Review · Reviewer_y8GR · 2025-03-11

**Overall Recommendation:** 4

**Summary:**

This work explores the role of diversity in adversarial ensemble learning, focusing on its definition and impact on algorithmic performance. The authors demonstrate that precisely calculating diversity is NP-Hard, distinguishing it from traditional diversity analysis. They introduce the first diversity decomposition in adversarial ensemble learning, decomposing adversarial ensemble loss into average of individual adversarial losses, prediction diversity, gradient diversity, and cross diversity—challenging previous methods that considered only gradient diversity. Extending this decomposition to classification with cross-entropy loss, they propose a novel ensemble method based on orthogonal adversarial predictions to enhance both gradient and cross diversity. Empirical results confirm the effectiveness of their approach.

## update after rebuttal
I have carefully read the authors' rebuttal and the feedback has well addressed my questions and concerns. Thus, I would like to insist on the score of accept . Thanks.

**Claims And Evidence:**

The claims made in the submission are supported by clear and convincing evidence.

1. The authors establish that calculating diversity precisely is NP-hard.

2. They introduce the first diversity decomposition in adversarial ensemble learning.

3. They propose a novel ensemble method based on orthogonal adversarial predictions to enhance the proposed diversity.

Evidences

1. The NP-hardness of the problem is demonstrated through a reduction from the 3-SAT problem, as proven in Theorem 3.1.

2. Theorems 3.2 and 3.3 present the diversity decomposition with respect to squared loss and cross-entropy loss, respectively. Experimental results further validate the correctness of the decomposition, as shown in Figures 1 and 3.

3. Empirical results confirm that the proposed method enhances diversity (Figure 7) and improves robustness (Table 1).

**Essential References Not Discussed:**

To the best of my knowledge, this paper cites all essential works related to traditional diversity or adversarial diversity.

**Experimental Designs Or Analyses:**

I have checked the experiments and found them to be sound and valid.

The experiments in Section 3 validate the correctness of the proposed diversity measures and highlight the limitations of previous approaches. Furthermore, the experiments in Section 5 confirm the effectiveness of the proposed method, demonstrating increased diversity (Figure 7) and enhanced robustness (Table 1).

**Methods And Evaluation Criteria:**

The proposed diversities and ensemble method make sense for adversarial ensemble learning.

The proposed diversity measures are derived through decomposition, clearly illustrating their impact on adversarial ensemble loss. As demonstrated in Theorems 3.2 and 3.3, greater diversity leads to a lower adversarial ensemble loss.

Building on this insight, the authors emphasize the orthogonality of adversarial predictions. Empirical results further validate the effectiveness of the proposed method, showing enhanced diversity (Figure 7) and improved robustness (Table 1).

**Other Comments Or Suggestions:**

1. The author has scattered the introduction of related work throughout the main text. It would be better to add a “Related Work” section in the appendix to summarize previous studies.

2. It would be better to investigate how the hyper-parameters affect the performance of the proposed method.

3. It would be better to conduct experiments on other l_p norms.

**Other Strengths And Weaknesses:**

Strengths:

1. This work is well-structured and easy to follow, with clear proofs and informative supplementary materials. The proof of NP-hardness in Theorem 1 constructs two neural networks by introducing an auxiliary variable, which is skillful.

2. The authors demonstrate that the proposed diversities can answer what the diversity is in adversarial ensemble learning, both theoretically and empirically. They also show that previous diversity measures may fail to capture true diversity, as illustrated in Example 1 and Figure 2.

3. The analysis of diversity in cross-entropy loss suggests that attention should be given to the predictions of adversarial examples of base learners. This insight is both natural and valuable, and experimental results further validate it.

4. Extensive experiments are conducted, demonstrating the effectiveness of the proposed method. As shown in Figure 7, the proposed method enhances the proposed diversities, leading to improved adversarial accuracy.

Weaknesses:

1. This work proposes the diversity with the first-order approximation, which is somewhat limited. The neural network may not exhibit strong linearity in local regions.

2. The proof of the decompositions is relatively simple and does not involve any special techniques.

**Questions For Authors:**

1. What is the time complexity of Algorithm 1?

2. Can the decomposition be applied to other types of loss functions?

**Relation To Broader Scientific Literature:**

This work defines the diversity for adversarial ensemble by decomposing the adversarial ensemble loss to the average of individual losses and diversities. While such decomposition has been explored in non-adversarial environment [1,2], this work introduces the first decomposition in adversarial environment.

Previous works define the diversity via gradient [3,4]. The proposed decomposition shows that it is not sufficient to only consider gradient to characterize the diversity.

[1] Zhou, Z.-H. Ensemble Methods: Foundations and Algorithms. CRC Press, 2012.

[2] Wood, D., Mu, T., Webb, A. M., Reeve, H. W., Lujan, M., and Brown, G. A unified theory of diversity in ensemble learning. Journal of Machine Learning Research, 24 (359):1–49, 2023.

[3] Dabouei, A., Soleymani, S., Taherkhani, F., Dawson, J., and Nasrabadi, N. M. Exploiting joint robustness to adversarial perturbations. In Proceedings of the IEEE/CVF Conference on Computer Vision and Pattern Recognition, pp. 1122–1131, Seattle, WA, 2020.

[4] Huang, B., Ke, Z., Wang, Y., Wang, W., Shen, L., and Liu, F. Adversarial defence by diversified simultaneous training of deep ensembles. In Proceedings of the 35th AAAI Conference on Artificial Intelligence, pp. 7823–7831, Virtual Event, 2021.

**Theoretical Claims:**

I have reviewed all the proofs, including the NP-hardness in Theorem 3.1, the diversity decompositions in Theorem 3.2 and 3.4, and others.

The NP-hardness is demonstrated through a reduction from the 3-SAT problem. The authors build two neural networks given a 3-SAT problem and show the value of the diversity of the neural networks can determine the satisfiability of the 3-SAT problem. The proof of the decomposition is relatively simple, mainly involving some equation transformation techniques and solving for adversarial perturbations.

I found no errors.

---

> ### Author Rebuttal · Authors · 2025-04-01
>
> [Q1] This work proposes the diversity with the first-order approximation, which is somewhat limited. The neural network may not exhibit strong linearity in local regions.
>
> [A1] We will clarify that the first-order approximation is motivated from previous first-order approximation methods for robust learning [Kariyappa & Qureshi 2019; Dabouei et al. 2020; Huang et al. 2021; Bogun et al., 2022], and it is difficult to make similar analysis for higher-order (even for second-order) approximations without closed-form solutions of adversarial examples and loss functions, which is relevant to the roots of polynomials of $2d$ degree ($d$ is the input dimensionality) [More & Sorensen, 1983; Fortin & Wolkowicz, 2004].
>
> [Q2] What is the time complexity of Algorithm 1?
>
> [A2] We will clarify that the time complexity of Algorithm 1 takes $m$-times as that of training a single neural network ($m$ is the number of neural networks in the ensemble). In addition, it takes $O(m^3)$ computational cost for the regularization with its gradients.
>
> [Q3] Can the decomposition be applied to other types of loss functions?
>
> [A3] We will clarify that this work makes the decompositions for two commonly-used loss functions, squared loss for regression and cross-entropy loss for neural network in classification, and it can be applied to other loss functions such as Poisson loss as in [Wood et al., 2023].

---

> > ### Comment · Reviewer_y8GR · 2025-04-05
> >
> > I have carefully read the authors' rebuttal and the feedback has well addressed my questions and concerns. Thus, I would like to insist on the score of 4 (accept) . Thanks.

---

### Official Review · Reviewer_wKSP · 2025-03-12

**Overall Recommendation:** 4

**Summary:**

This paper addresses the critical role of diversity in adversarial ensemble learning, demonstrating that precisely calculating diversity for neural networks under adversarial perturbations is NP-hard due to structural and predictive interdependencies. By decomposing adversarial ensemble loss into four components—average individual losses, prediction diversity, gradient diversity, and cross diversity—the authors reveal the insufficiency of prior gradient-centric approaches. They propose AdvEOAP, a method that enforces orthogonal adversarial predictions through KL divergence regularization and entropy maximization, simultaneously improving gradient and cross diversity. Empirical validation on MNIST, F-MNIST, and CIFAR10 shows state-of-the-art robustness, with AdvEOAP outperforming baselines by up to 80% under AutoAttack, while ablation studies confirm the necessity of diversity components. The work establishes a theoretical foundation for adversarial ensemble diversity and offers a practical framework for enhanced robustness.

**Claims And Evidence:**

The claims in the paper are supported by rigorous theoretical and empirical evidence. The NP-hardness of calculating adversarial ensemble diversity is proven via a reduction from the 3-SAT problem, demonstrating that precise diversity computation requires solving an NP-hard optimization due to structural and predictive interdependencies. The proposed adversarial loss decomposition (into average individual losses, prediction/gradient/cross diversity) is validated theoretically through first-order Taylor approximations (Theorems 3.2 and 3.4) and empirically via experiments on MNIST, F-MNIST, and CIFAR10, showing gradient/cross diversity dominate under large perturbations. The effectiveness of AdvEOAP is confirmed by ablation studies (e.g., 1.1–1.5% accuracy drop without orthogonality regularization) and superior robustness over baselines (e.g., +40% accuracy vs. DVERGE under AutoAttack). Critiques of prior gradient-centric methods are supported by constructed counterexamples (Example 1) and empirical trends (Figure 2), where adversarial loss decreases despite unchanged gradient alignment. While the analysis relies on first-order approximations and scalability to larger models remains unexplored, the core claims are substantiated by theoretical proofs, controlled experiments, and statistically significant results across 50 runs.

**Essential References Not Discussed:**

N/A

**Experimental Designs Or Analyses:**

The experimental designs and analyses are methodologically sound but have minor limitations. The authors evaluate AdvEOAP on MNIST, F-MNIST, and CIFAR-10—standard benchmarks for adversarial robustness—using diverse attacks (FGSM, PGD, AutoAttack) to cover gradient-based, iterative, and adaptive threat models. The inclusion of ablation studies (Table 2) effectively isolates the impact of orthogonality regularization, showing a 1.1–1.5% accuracy drop when removed. Statistical rigor is ensured via 50 runs with reported mean ± std. deviations, and Figure 7’s training dynamics validate the theoretical link between diversity and robustness. However:

1. Scalability gaps: Experiments are limited to ResNet20 on small/medium datasets; larger architectures (e.g., Vision Transformers) and high-resolution datasets (e.g., ImageNet) are untested, leaving scalability in doubt.

2. Attack parameter details: While AutoAttack is included, hyperparameters for weaker attacks (e.g., FGSM step size) are not fully disclosed, limiting reproducibility.

3. Computational costs: Training time and resource requirements for AdvEOAP (e.g., orthogonalization overhead) are unquantified, which is critical for real-world deployment.

Overall, the experiments robustly validate the method’s core claims but leave practical scalability and efficiency as open questions.

**Methods And Evaluation Criteria:**

The proposed methods and evaluation criteria are aligned with the problem of adversarial ensemble learning. The AdvEOAP method, which enforces orthogonal adversarial predictions via KL divergence and determinant-based regularization, directly addresses the need for diverse vulnerability patterns among base learners-a critical requirement for robustness against adaptive attacks. By optimizing both gradient diversity and cross diversity, the method systematically tackles the limitations of prior gradient-centric approaches. The evaluation criteria are appropriate and comprehensive.

**Other Comments Or Suggestions:**

N/A

**Other Strengths And Weaknesses:**

Enter any comments on other strengths and weaknesses of the paper, such as those concerning originality, significance, and clarity. We encourage you to be open-minded in terms of potential strengths. For example, originality may arise from creative combinations of existing ideas, removing restrictive assumptions from prior theoretical results, or application to a real-world use case (particularly for application-driven ML papers, indicated in the flag above and described in the Reviewer Instructions).

**Questions For Authors:**

Q1. Theoretical Generality: The NP-hardness proof (Theorem 3.1) assumes ReLU activations and specific perturbation bounds. How generalizable are these results to other activation functions (e.g., sigmoid) or arbitrary perturbation norms?

Q2. First-Order Approximation Limits: The loss decomposition (Theorems 3.2/3.4) relies on first-order Taylor expansions. How might higher-order terms affect the validity of the decomposition under large perturbations (e.g., ϵ>0.1) or highly nonlinear regimes?

Q3. Scalability: Experiments are limited to ResNet20 on small/medium datasets (MNIST, CIFAR-10). Can AdvEOAP scale to larger architectures (e.g., Vision Transformers) or high-resolution datasets (e.g., ImageNet) without prohibitive computational costs?

Q4. Computational Overhead: Training time and resource requirements for AdvEOAP need to be quantified in theory and experiments.

Q5. Attack Parameter Transparency: Hyperparameters for comparison methods are not fully disclosed. This might affect reproducibility and comparative analysis.

**Relation To Broader Scientific Literature:**

The paper’s contributions significantly advance the adversarial robustness literature by addressing critical gaps in ensemble diversity theory. Prior works (e.g., Kariyappa & Qureshi, 2019; Dabouei et al., 2020) focused on gradient misalignment as the primary diversity metric but lacked theoretical grounding for its sufficiency. This work bridges that gap by proving that diversity in adversarial settings is inherently NP-Hard (extending Katz et al.’s (2017) NP-Hardness results for neural network verification) and introducing a multi-component diversity decomposition—building on classical ensemble theories (Krogh & Vedelsby, 1994; Zhou, 2012) but adapting them to adversarial perturbations. The AdvEOAP method’s orthogonality regularization draws inspiration from Pang et al. (2019), who used prediction disagreements, but innovates by combining KL divergence and determinant-based constraints to enforce structural diversity. These contributions align with broader trends in robustness research (e.g., Yang et al.) that emphasize diversifying vulnerability patterns but provide the first formal framework linking diversity components to adversarial loss. By unifying theoretical rigor (NP-Hardness, decomposition) with practical algorithm design, the paper addresses a longstanding challenge in adversarial ensemble learning: how to systematically define and optimize diversity for robustness.

**Theoretical Claims:**

The theoretical claims in the paper are correct under their stated assumptions but face limitations in generality. The NP-hardness proof for diversity calculation (Theorem 3.1) is valid for ReLU networks under specific perturbation bounds but does not generalize to all activation functions or norms. The loss decomposition theorems (3.2 and 3.4) rigorously hold under first-order Taylor approximations but neglect higher-order effects, limiting applicability to small perturbations (ϵ) or highly nonlinear regimes. Overall, the core proofs are logically consistent but rely on simplifications (e.g., linearized adversarial perturbations) that may not fully capture real-world model behaviors, particularly for deep networks with complex interactions.

---

> ### Author Rebuttal · Authors · 2025-04-01
>
> [Q1] …The NP-hardness proof (Theorem 3.1) assumes ReLU activations and specific perturbation bounds. How generalizable are these results to other activation functions (e.g., sigmoid) or arbitrary perturbation norms?
>
> [A1] We will clarify that our results hold for $l_p$ norms with $p = 1, 2, \cdots, \infty$ in Theorem 3.1, and it is feasible to generalize to other activation functions (e.g., sigmoid) based on universal approximation [Kidger et al. 2020], i.e., a 3-SAT problem can be reduced to a diversity problem of ReLU neural networks as in Theorem 3.1, and network employing other activation functions (e.g., sigmoid) can approximate any ReLU network with arbitrary precision.
>
> [Q2] First-Order Approximation Limits: The loss decomposition (Theorems 3.2/3.4) relies on first-order Taylor expansions. How might higher-order terms affect the validity of the decomposition…
>
> [A2] We will clarify that the first-order approximation is motivated from previous first-order approximation methods for robust learning [Kariyappa & Qureshi 2019; Dabouei et al. 2020; Huang et al. 2021; Bogun et al., 2022], and it is difficult to make similar analysis for higher-order (even for second-order) approximations without closed-form solutions of adversarial examples and loss functions, which is relevant to the roots of polynomials of $2d$ degree ($d$ is the input dimensionality) [More & Sorensen, 1983; Fortin & Wolkowicz, 2004].
>
> [Q3] Scalability: Experiments are limited to ResNet20 on small/medium datasets (MNIST, CIFAR-10). Can AdvEOAP scale to larger architectures (e.g., Vision Transformers) or high-resolution datasets (e.g., ImageNet) without prohibitive computational costs?
>
> [A3] We will clarify that we consider ResNet20 on datasets Minist, F-minist and Cifar10 for fair comparisons with previous ensemble methods, and we have been conducting experiments on larger datasets with complex network architectures. Here are some preliminary results:
>
> Adversarial accuracy for CIFAR100 with 5 ResNet18
>
> | Methods | PGD20 | AutoAttack |
> |-|-|-|
> | Our Method | 25.16 | 24.39 |
> | AdvADP | 23.690 | 21.22 |
>
> We could present more experimental results in the following days due to computational resources. For example, it is about 4 days to train a deep neural network on Tiny-ImageNet with four A6000 GPUs, while we only have a single NVIDIA GeForce RTX 4090 GPU to train multiple deep neural networks for diversity.
>
> [Q4] Computational Overhead: Training time and resource requirements for AdvEOAP need to be quantified in theory and experiments.
>
> [A4] We will clarify that, for AdvEOAP, the training time takes $m$-times as that of training a single neural network ($m$ is the number of neural networks in ensemble). In addition, it takes $O(m^3)$ computational cost for the regularization with its gradients. For resource requirements, AdvEOAP needs $m$ times the space complexity of a single neural network.
>
> Here are some experiments as follows:
>
> For MNIST dataset:
>
> | Methods | GAL | ADP | AdvADP | DVERGE | PDD | TRS | $\text{iGAT}_{\text{ADP}}$ | Our $\text{AdvE}_{\text{OAP}}$ |
> |-|-|-|-|-|-|-|-|-|
> | Time per Epoch (s) | 23.45 | 7.14 | 45.36 | 40.65 | 70.72 | 241.8 | 22.93 | 41.45 |
>
> For F-MNIST dataset:
>
> | Methods | GAL | ADP | AdvADP | DVERGE | PDD | TRS | $\text{iGAT}_{\text{ADP}}$ | Our $\text{AdvE}_{\text{OAP}}$ |
> |-|-|-|-|-|-|-|-|-|
> | Time per Epoch (s) | 23.45 | 7.15 | 45.50 | 39.52 | 69.48 | 219.32 | 23.94 | 41.32 |
>
> For CIFAR10 dataset:
>
> | Methods | GAL | ADP | AdvADP | DVERGE | PDD | TRS | $\text{iGAT}_{\text{ADP}}$ | Our $\text{AdvE}_{\text{OAP}}$ |
> |-|-|-|-|-|-|-|-|-|
> | Time per Epoch (s) | 69.47 | 15.18 | 134.54 | 172.93 | 291.07 | 626.33 | 100.51 | 125.48 |
>
> All methods require approximately 4.4GB memory except TRS with 10.3GB memory due to storing the Hessian matrix.
>
> [Q5] Parameter details: Hyperparameters for comparison methods are not fully disclosed. While AutoAttack is included, hyperparameters for weaker attacks (e.g., FGSM step size) are not fully disclosed, limiting reproducibility.
>
> [A5] We will clarify more parameter details in Appendix: For the FGSM, PGD10, and PGD20 attack methods, the hyperparameters are set as
>
> | Method | Steps | Step Size |
> |-|-|-|
> | FGSM | 1 | $\epsilon$ |
> | PGD10 | 10 | $\epsilon/3$ |
> | PGD20 | 20 | $\epsilon/6$ |
>
> Here, $\epsilon$ is the perturbation size.
> For the AutoPGD, MORA, and AutoAttack attack methods, we set the number of steps to 20, and keep all other parameters in the original reference or the provided code defaults.
>
> We will add a formal description for our algorithm in Appendix to clarify orthogonalization implementation, and release the code to ensure reproducibility.

---

### Official Review · Reviewer_gZFG · 2025-03-14

**Overall Recommendation:** 3

**Summary:**

This paper considers the problem of defining and correlating diversity with algorithmic performance in adversarial ensemble learning. The authors first prove that precisely calculating the diversity in adversarial ensemble learning is an NP-Hard problem. They then propose a new diversity decomposition for adversarial ensemble learning, breaking down the adversarial ensemble loss into four components: the average of individual adversarial losses, prediction diversity, gradient diversity, and cross diversity. This decomposition highlights the insufficiency of considering only gradient diversity, as done in prior works. The authors also extend this decomposition to classification tasks using cross-entropy loss. Based on their theoretical analysis, they develop a new ensemble method called AdvE$_{oap}$, which improves gradient and cross diversity simultaneously by orthogonalizing adversarial predictions of base learners.

After rebuttal:

The authors have address my questions. I will keep my score.

**Claims And Evidence:**

In general the Claims are clear. The claims made in the paper are well-supported by both theoretical analysis and empirical evidence.

Yet I think the following discussions could be improved.

1) There is a gap between NP-hard problem to first-order approximation. It is a very weak argument that we only consider first order approximation since the complete analysis is a NP-hard problem. There are still many things we can do in between.

2) In the theorem of decomposition (Theorem 3.2 and 3.4), It is weird to me to you equally, since those are only first order approximation

**Essential References Not Discussed:**

To my knowledge, no.

**Experimental Designs Or Analyses:**

The experimental design is appropriate for validating the theoretical claims.

After the authors proposed the decomposition, they provide experiments to see which components are more relevant. This helps to justify their propose method AdvE$_{oap}$.

Then, the authors conduct extensive experiments on three datasets (MNIST, F-MNIST, and CIFAR10) and compare their method against several state-of-the-art adversarial ensemble methods. The use of multiple adversarial attacks (PGD, AutoPGD, Autoattack etc.) ensures that the evaluation is comprehensive.

**Methods And Evaluation Criteria:**

The proposed methods are well-suited for the problem of adversarial ensemble learning. The authors use first-order Taylor approximation to decompose the adversarial ensemble loss. The evaluation criteria include classification accuracy under various adversarial attacks (PGD, AutoPGD, AutoAttack etc.), which are standard benchmarks in adversarial robustness research. The experiments are conducted on widely used datasets, and the results are compared against several baseline methods, ensuring a comprehensive evaluation.

It would be better if the authors could provide experiments on larger dataset, such as TinyImageNet.

**Other Comments Or Suggestions:**

no

**Other Strengths And Weaknesses:**

Weaknesses:

The paper could benefit from a more detailed discussion of the limitations of the proposed method, particularly in scenarios where the assumptions of first-order approximation may not hold.

**Questions For Authors:**

See above.

**Relation To Broader Scientific Literature:**

To my knowledge, no.

**Theoretical Claims:**

The theoretical claims are sound and well-justified. The authors provide detailed proofs for the NP-Hardness of calculating diversity (Theorem 3.1) and the diversity decomposition for both squared loss (Theorem 3.2) and cross-entropy loss (Theorem 3.4). The proofs are rigorous.

Again, as I mention before, it is weird to me to see equality when using first order approximation.

---

> ### Author Rebuttal · Authors · 2025-04-01
>
> [Q1] … a gap between NP-hard problem to first-order approximation …. only consider first order approximation since the complete analysis is a NP-hard problem.
>
> [A1] We will clarify that the first-order approximation is motivated from previous first-order approximation methods for robust learning [Kariyappa & Qureshi 2019; Dabouei et al. 2020; Huang et al. 2021; Bogun et al., 2022], and it is difficult to make similar analysis for higher-order (even for second-order) approximations without closed-form solutions of adversarial examples and loss functions, which is relevant to the roots of polynomials of $2d$ degree ($d$ is the input dimensionality) [More & Sorensen, 1983; Fortin & Wolkowicz, 2004].
>
>
> [Q2] In the theorem of decomposition (Theorem 3.2 and 3.4), It is weird to me to you equally, since those are only first order approximation.
>
> [A2] We will clarify that we consider the first-order approximation in the decomposition theorems, and we will take the approximately equal rather than exactly equal.
>
> [Q3] It would be better if the authors could provide experiments on larger dataset, such as TinyImageNet.
>
> [A3] We will clarify that we consider datasets Minist, F-minist and Cifar10 for fair comparisons with previous ensemble methods, and we have been conducting experiments on larger datasets with complex network architectures. Here are some preliminary results:
>
> Adversarial accuracy for CIFAR100 with 5 ResNet18
>
> | Methods | PGD20 | AutoAttack |
> |-|-|-|
> | Our Method | 25.16 | 24.39 |
> | AdvADP | 23.690 | 21.22 |
>
> We could present more experimental results in the following days due to computational resources. For example, it is about 4 days to train a deep neural network on Tiny-ImageNet with four A6000 GPUs, while we only have a single NVIDIA GeForce RTX 4090 GPU to train multiple deep neural networks for diversity.
>
> [Q4] The paper could benefit from a more detailed discussion of the limitations of the proposed method, particularly in scenarios where the assumptions of first-order approximation may not hold.
>
> [A4] We will add a figure to discuss the limitations of the proposed method with and without first-order approximations, and the basic idea is to consider the residual terms on the first-order approximation in experiments and analyze its influence.

---

### Decision · Program_Chairs · 2025-05-01

**Decision:**

Accept (poster)

**Comment:**

This paper introduce an ensemble method that enhances robustness by orthogonalizing adversarial predictions to increase both gradient and cross diversity.

Strengths consistently noted by reviewers include:
- Reviewer wKSP and Reviewer y8GR praised the novelty and theoretical rigor of the NP-Hardness proof and diversity decomposition.
- Reviewer gZFG appreciated the sound experimental method and impact of orthogonal prediction regularization.
- Reviewer ybcr highlighted the technical depth, clarity, and robustness improvements over prior adversarial ensemble methods.

Reviewers acknowledged and appreciated the authors’ responses. Before the rebuttal, Reviewers raised concerns regarding the limitations of relying on first-order approximations in theoretical decomposition, the lack of experiments on larger-scale datasets and architectures, the absence of analysis under advanced attack settings like EOT and BPDA. Major concerns were thoroughly addressed in the rebuttal with additional theoretical clarifications, new empirical results under EOT/BPDA, training time comparisons, and scalability discussions.

Therefore, AC recommends acceptance of the paper.